



# Estimating propagation probability from meteorological to ecological droughts using a hybrid machine learning-Copula method

Tianliang Jiang[1,2], Xiaoling Su[1,2], Gengxi Zhang[3], Te Zhang[1,2], Haijiang Wu[1,2]

[1] College of Water Resources and Architectural Engineering, Northwest A & F University, Yangling, Shaanxi, 712100, China

[2] Key Laboratory for Agricultural Soil and Water Engineering in Arid Area of Ministry of Education, Northwest A & F University, Yangling, Shaanxi, 712100, China

[3] College of Hydraulic Science and Engineering, Yangzhou University, Yangzhou 225012, China

*Correspondence to*: Xiaoling Su (xiaolingsu@nwafu.edu.cn)

**Abstract.** The impact of droughts on vegetation is essentially manifested as the transition of water shortage from the meteorological to ecological stages. Therefore, understanding the mechanism of drought propagation from meteorological to ecological drought is crucial for ecological conservation. This study proposes a method for calculating the probability of meteorological drought to trigger ecological drought at different magnitudes in Northwestern China. In this approach, meteorological and ecological drought events during 1982–2020 are identified using the three-dimensional identification method; the propagated drought events are extracted according to a certain spatio-temporal overlap rule; and propagation probability is calculated by coupling machine learning model and C-vine copula. The results indicate that: (1) 46 drought events are successfully paired by 130 meteorological and 184 ecological drought events during 1982–2020; ecological drought exhibits a longer duration, but smaller affected area and severity than meteorological drought; (2) Quadratic Discriminant Analysis (QDA) classifier performs the best among the 11 commonly used machine learning models which is combined with four-dimensional C-vine copula to construct drought propagation probability model; (3) the hybrid method considers more drought characteristics and more detailed propagation process which addresses the limited applicability of the traditional method to regions with large spatial extent.

## 1 Introduction

Drought is a multivariable and complex natural hazard with the characteristics of slow evolution, wide impact, and spatial extent (Feng et al., 2021; Wu et al., 2021; Zhang et al., 2021a; Zhang et al., 2021b). Conventionally, drought can be classified into meteorological drought, hydrological drought, agricultural drought, and socioeconomic drought. It is commonly accepted that all types of drought originate from meteorological drought (Mishra and Singh, 2010). Crausbay et al (2017) argued that existing drought types are described through a "human-centric" lens to characterize a range of effects generated by meteorological drought. This implies that the response of ecosystem to drought are generally ignored in policy development, which in turn elicit water use conflicts between humans and ecosystems (Zhang et al., 2021c). Ecological Drought Working Group of Science for Nature and People Partnership (SNAPP) proposed a framework of ecological drought from an "ecology-centric" lens, which incorporates ecological, meteorological, and hydrological information (Crausbay et al., 2017). Ecological



drought was thus defined as an episodic deficit in water availability that drives ecosystems beyond thresholds of resilience into a vulnerable state, impacts ecosystem services, and triggers feedbacks in natural and/or human systems (Bradford et al., 2020; Crausbay et al., 2017; McEvoy et al., 2018; Munson et al., 2021; Raheem et al., 2019).

Vegetation is among the most important components in terrestrial ecosystems, and the distribution and growth of vegetation are largely influenced by meteorological factors (Wang et al., 2021; Zeng et al., 2020; Zhang et al., 2021d). Developments in remote sensing technology have facilitated the application of vegetation indices to reflect the response of vegetation to climate change (Lawal et al., 2021). For example, a simple linear relationship was found between the standardized precipitation evapotranspiration index (SPEI) and normalized difference vegetation index (NDVI) at the global scale (Vicente-Serrano et

al., 2012). The correlation between SPEI and NDVI showed a positive relationship in most regions of northwestern China (NWC), with the exception of a few regions such as the western parts of the Tarim Basin, Qaidam Basin, and southeastern part of the area (Jiang et al., 2018). Actually, the impact of drought on vegetation is manifested as the transition of water shortage from the meteorological stage to ecological stage. Therefore, this impact should be analyzed by quantifying the effect of decreasing precipitation on the variation of available ecological water, i.e., from the perspective of drought propagation.

Drought propagation refers to the transition of one drought type to another, and it is vital for drought monitoring and prediction (Fang et al., 2020; Warter et al., 2021). Accordingly, drought propagation has become a hot topic in meteorology and hydrology fields (Apurv et al., 2017; Guo et al., 2020). Approaches to drought propagation analysis are broadly divided into model simulations and statistical methods (Han et al., 2019). In the former approach, hydrological responses to meteorological drought are analyzed by using physical based models that are considered to be effective in representing relevant hydrological

processes (Yu et al., 1999). Nevertheless, this approach involves labor-intensive calibration processes and is not suitable at large spatial scales (Huang et al., 2017). In contrast, statistical methods with fewer assumptions are easier to use at different spatial scales (Huang et al., 2017). However, in such methods, the propagation process was analyzed using the time series of an average value of drought index in a region or subregion (explained in the Discussion section). In other words, they considered temporal connection of two drought types and ignored their spatial overlap, which may result in the miscalculation

of drought propagation in regions with large spatial extent.

Probabilistic model has been proved to be a better way to quantify the relationship between different types of droughts (Ayantobo et al., 2018; Chang et al., 2016; Das et al., 2020). This is due to that the probability information of one type of successive drought events is contained in another type of drought associated with it (Wu et al., 2021). For example, Guo et al (2020) calculated occurrence probability of hydrological drought based on different intervals of duration and severity of

meteorological drought. Sattar et al (2019) identified the occurrence probability of different classes and lag time of hydrological drought according to intensity of meteorological drought. Nevertheless, the number of drought characteristics considered in these studies are relatively few. Xu et al. (2015a) found that drought occurrence probability would be underestimated if fewer drought characteristics were considered. Therefore, the traditional drought probabilistic model of drought propagation can be improved by introducing the three-dimensional drought identification method, which provides

more drought information (Liu et al., 2019).



Taking a typically ecological fragile region, Northwestern China (NWC), as an example, the motivation of this study is to identify meteorological drought and ecological drought during 1982–2020 in the NWC from a three-dimensional perspective, and propose a novel method to investigate the response probability of ecological drought to meteorological drought. The remainder of the current paper is organized as follows: Section 2 briefly overviews the geographic information of NWC.

Section 3 describes the datasets used in this paper, and the procedure for estimating propagation probability from meteorological to ecological drought. The results and the comprehensive analysis of the proposed approach are presented in section 4 and 5, respectively. Finally, the conclusions are given in section 6.

## 2 Study area

Northwestern China (NWC; 31°35′N–49°15′N, 73°25′E–111°15′E) includes the provinces of Shaanxi, Gansu, Qinghai, and the

Autonomous Regions of Xinjiang Uyghur and Ningxia Hui, covering a total surface area of 3.1 million km$^2$ (Fig. 1) (Zheng et al., 2021). The terrain of NWC constitutes mountains, basins, and the Gobi. The altitude ranges from –156 m to 6647 m, showing the characteristics of "west high and east low". Four climatic divisions, including humid, semi-humid, semi-arid, and arid area were demarcated, based on the dryness index (Zhang et al., 2021d). As NWC is located at the upstream of the Yangtze, Yellow and other large rivers, it is significant to study the impact of drought on its ecosystem (Liu et al., 2021).

## 80    3 Materials and methods

### 3.1 Datasets

Monthly meteorological data, including surface reflectance, temperature, relative humidity, atmospheric pressure, downward shortwave radiation, wind speed, and longwave radiation, obtained from the ERA5-land reanalysis dataset (https://cds.climate.copernicus.eu) issued by European Centre for Medium-Range Weather Forecasts (ECWMF), which has a

spatial resolution of 0.1° × 0.1° and covers the period of 1981–2021. Root soil moisture data were obtained from the hydrological dataset, simulated by Noah model of the Global Land Data Assimilation System (GLDAS, 0.25°×0.25°; https://ldas.gsfc.nasa.gov/gldas), covering the period of 1948–2021. NDVI data covering the period 1981–2021 were obtained from the National Centers for Environmental Information (NCEI) (https://www.ncei.noaa.gov/), with a spatial resolution of 0.05° × 0.05°. Land use type data (LUTD) with a spatial resolution 1 km was downloaded from China's multi-period land

use/cover change monitoring dataset (http://www.resdc.cn); it includes years of 1980, 1990, 1995, 2000, 2005, 2010, 2015, 2018, and 2020. In order to uniform the spatial resolution of Root soil moisture, all spatial datasets were resampled to 0.25°×0.25° using the bilinear interpolation method. The temporal range of all datasets were extracted from January 1982 to December 2020.



## 3.2 Meteorological and ecological drought index

Previous studies found that the standardized precipitation evaporation index (SPEI) overestimated the meteorological drought in NWC where actual atmospheric water demand is determined by precipitation variation (Ayantobo and Wei, 2019; Zhang et al., 2019a; Zhang et al., 2021b). Additionally, precipitation is the main water resources for vegetation growth in most regions of NWC due to the deep phreatic buried depth (Cao et al., 2021). Standardized precipitation index (SPI) was thus used in the current study to represent meteorological drought. SPI at different time scales was calculated by aggregating $n$-month moving

sums (McKee et al., 1993). For example, SPI–3 in March was calculated by accumulating the series of precipitation in January, February, and March. SPI–3 has been reported to be highly representative of the impacts of meteorological conditions on vegetation as the vegetation variation is sensitive to three months accumulated precipitation (McKee et al., 1993; Vicente-Serrano et al., 2012; Vicente-Serrano et al., 2010; Vicenteserrano et al., 2010). Therefore, SPI-3 was used to characterize meteorological drought in this study. Further details on SPI calculation are available in (McKee et al., 1993).

Commonly used drought indices indirectly reflect the influence of drought on ecosystems, and they do not comprehensively reflect the homeostasis between ecological water consumption and requirement in drought evolution (Jiang et al., 2021). Additionally, decreases in vegetation coverage are not only caused by a persistent deficit in available water for ecosystems but also other aspects, such as wildfire, hail, flood, and human activities (Bento et al., 2020). This limited the ability of vegetation indices to reflect drought conditions. Therefore, a new drought index, the standardized ecological water deficit index (SEWDI),

was constructed to monitor terrestrial ecological drought in our previous study (Jiang et al., 2021). SEWDI follows a similar procedure as SPI. Ecological water deficit (EWD) is the difference between FAO-based ecological water requirement and SEBS-based ecological water consumption (Chi et al., 2018; Jiang et al., 2021). Therefore, SEWDI reflects the dynamics of energy and water balance under human activities and climate change. Additionally, the standardization method facilitates the same threshold and evaluation criteria in monitoring two drought types (Peng et al., 2019; Zang et al., 2020), which reduces

the influence of other algorithms on final results and guarantees spatio-temporal comparability (Liu et al., 2017). The procedure in calculating SEWDI calculation is detailed in Jiang et al (2021).

## 3.3 Drought propagation probability method

Since a reliable understanding of the drought propagation process is beneficial for drought forecasting, research interest in the probability of drought propagation from meteorological droughts to other types of droughts has been increasing (Zhou et al.,

2021). The current study thus proposed a novel method coupling spatial and temporal connection method of two type droughts, with machine learning model, and C-vine copula to investigate the relationship between meteorological and ecological drought. A flow diagram of the method is depicted in Fig. 2.

Investigating the relationship between the characteristics of the two drought types is key to constructing a probability model. The approach is summarized in two steps as follows:



Step 1: Meteorological and ecological drought events were identified from a three-dimensional perspective, respectively (Section 3.3.1).

Step2: The two drought types were paired on the basis of a certain spatio-temporal matching rule to extract propagated drought events. Their drought characteristics, including drought affected area, drought severity, and drought duration, were calculated according to the method described in Section 3.3.2.

Step 3: Taking the characteristics of meteorological drought extracted in step 2 as inputs, and propagation results as outputs, the optimal model was selected from 11 machine learning classification models to calculate the propagated probability (P1) of meteorological drought (Section 3.3.3). Then, a conditional probability model of the paired meteorological and ecological drought events was constructed based on C-vine copula (Section 3.3.4). According to the severities of all identified ecological drought events, cumulative probabilities of 0.5, 0.75, and 0.9 were selected to demarcate moderate, serious, and extreme

drought, respectively (Guo et al., 2020). The probabilities of ecological drought at different magnitudes triggered by meteorological drought were obtained by multiplying P1 with their conditional probability

### 3.3.1 Three-dimensional drought identification

According to Andreadis et al (2005), the evolution of a drought event should be viewed as a spatio-temporal continuum (longitude, latitude, and time). Different from traditional one- or two-dimensional drought identification method, the three-

dimensional array of SPI-3 and SEWDI-3 were extracted to characterize the degree of meteorological and ecological droughts. The extraction procedure involves two steps (Fig. 2) (Andreadis et al., 2005; Xu et al., 2015a; Xu et al., 2015b): firstly, the clustering method was used to identify drought patches in each month; secondly, the drought continuum was constructed by selecting the overlapping areas of drought patches between two adjacent months, which was greater than 1.6% of the total area (explained in the Discussion section).

For each drought event, three drought characteristics were extracted as follows: (1) affected area was calculated by cumulating the area affected by drought in each month during the entire drought period. (2) Duration denotes the time of a drought event persisted. (3) Severity is a cumulative value of SEWDI-3 or SPI-3 for the entire drought duration and areal extent, and equals to the volume of the three-dimensional continuum.

### 3.3.2 Spatio-temporal connection of two drought types

Liu et al (2019) developed a new method for identifying the propagation between two related drought types based on the drought identification method from a three-dimensional perspective. The current study employed this method to identify the propagation from meteorological to ecological drought. The key to this method is the determination of the temporal and spatial connection between two drought types. The specific steps are as follows.

Firstly, the identified meteorological and ecological drought events are sorted in the chronological order. Secondly, whether

the two drought types overlap in time is judged according to Eq. (1)–Eq. (2).



$$Overlap_t = \begin{cases} 1 \text{ if } \begin{cases} MBT_i \leq EBT_j \text{ and } \min\left(MET_i, EET_j\right) - \max\left(MBT_i, EBT_j\right) \geq 2 \\ MBT_i > EBT_j \text{ and } \min\left(MET_i, EET_j\right) - \max\left(MBT_i, EBT_j\right) \geq \alpha \end{cases} \\ 0 \text{ if } MBT_i \geq EBT_j \text{ and } \min\left(MET_i, EET_j\right) - \max\left(MBT_i, EBT_j\right) < \alpha \end{cases} \tag{1}$$

$$\alpha = \min\left(\frac{MDD_i}{3}, \frac{EDD_j}{3}\right) \tag{2}$$

where 1 and 0 denote the existence and absence of time overlap between two drought types, respectively; $MBT_i$ and $EBT_j$ represent the beginning time of the $i$-th meteorological and $j$-th ecological drought events, respectively. Similarly, $MET_i$ and $EET_j$ represent the end time of the $i$-th meteorological and $j$-th ecological drought events, respectively; $MDD_i$ and $EDD_j$ indicate the duration of the $i$-th meteorological and $j$-th ecological drought events, respectively.

Secondly, whether the meteorological and ecological drought patches connecting at spatial scale is judged according to Eq. (3) and Eq. (4)

$$Overlap_s = \begin{cases} 1 \text{ if } MDA_i \cap EDA_j \geq \beta \\ 0 \text{ if } MDA_i \cap EDA_j < \beta \end{cases} \tag{3}$$

$$\beta = \max\left(1.6\% \cdot A_{NWC}, \min\left(MDA_i, EDA_j\right) \cdot b\right) \tag{4}$$

where 1 and 0 denote the existence and absence of spatial overlap between two drought types; $A_{NWC}$ represents the total area of the NWC; $MDA_i$ and $EDA_j$ represent the projected area of the $i$-th meteorological and $j$-th ecological drought events, respectively. $b$ is set as 15% in the current study (See Discussion for the reason)

Thirdly, successfully matched drought events are encoded following the chronological order. Cells in Fig. 2 represent the relationship between preliminarily identified events of the two drought types. The propagation type from meteorological to ecological drought can be classified into four categories: one ecological drought event induced by one meteorological drought event (one–to–one), multiple ecological drought events induced by one meteorological drought event (one–to–many), one ecological drought event induced by multiple meteorological drought events (many–to–one), and multiple ecological drought events induced by multiple meteorological drought events (many–to–many). The codes of cells are identical if propagation type belong to one–to–many, many–to–one, and many–to–many.

Finally, the characteristics of meteorological and ecological drought events that belong to the same paired drought event are integrated, respectively. Among them, total duration is the difference between latest-ending and earliest-starting drought events; total affected area is the projected area of all individual drought events; total severity is the sum of severities of individual drought events.

### 3.3.3 Drought propagation identification based on machine learning model

The purpose of this part is to identify whether a meteorological drought event has the potential to trigger ecological drought. Eleven commonly used machine learning classification models, including k-neighbors classifier (KN) (Parzen, 1962), support





vector machine (SVM) classifier (Ben-Hur et al., 2000), Gaussian Process (GP) classifier (Chen et al., 2020), Decision Tree (DT) classifier (Quinlan, 1986), Multi-layer Perceptron (MP) classifier (Cybenko, 1989), AdaBoost (AB) classifier (Freund

and Schapire, 1997), Gaussian Naïve Bayes (GNB) (Chan T.F., 1982 ), Quadratic Discriminant Analysis (QDA) (Cover, 1965), Gradient Boosting (GB) classifier (Friedman, 2001), XGBoost (XGB) classifier (Chen and Guestrin, 2016), and Random Forest (RF) classifier (Pal, 2005), were employed for propagation judgement. Drought duration, severity, and affected area of meteorological drought were set as the model inputs (Fig. 2). 1 and 0 were set as model target which represent propagation occurrence and non-occurrence, respectively. The classifiers of each model were trained and validated using 5-fold cross-

validation. The classifier with the highest summation of accuracy, precision, recall, F1 score, and Matthews correlation coefficient was selected as the optimal model.

$$accuracy = \frac{TP+TN}{TP+TN+FP+FN} \tag{5}$$

$$precision = \frac{TP}{TP+FP} \tag{6}$$

$$recall = \frac{TP}{TP+FN} \tag{7}$$

$$F_1 \; score = \frac{2 \cdot TP}{2 \cdot TP+FP+FN} \tag{8}$$

$$MCC = \frac{TP \cdot TN - FP \cdot FN}{\sqrt{(TP+FP) \cdot (TP+FN) \cdot (TN+FP) \cdot (TN+FN)}} \tag{9}$$

where $TP$ and $FN$ represent actual positives that are correctly and wrongly predicted, respectively; $TN$ and $FP$ represent actual negatives are correctly and wrongly predicted, respectively.

### 3.3.4 Drought propagation probability model based on C-vine copula

Five univariate distributions, including Johnsonsb (Soukissian, 2013), Gamma (THOM, 1958), Exponential (Marshall and Olkin, 1967), Pearson III (Wallis and Wood, 1985), and Weibull distribution (Thoman et al., 1969), were used to fit affected area, duration, and severity of meteorological drought and severity of ecological drought. The optimal distribution was selected according to the goodness of fit (GOF), which was estimated with Kolmogorov–Smirnov (KS) test (Marsaglia et al., 2003) and Root Mean Square Error (RMSE).

Commonly used Copulas, including elliptical Copula (Guassian) and four Archimedean Copulas (Clayton, Gumbel, Frank, and Joe), were used to join two marginal distributions (Chang et al., 2016). The GOF of these Copulas was estimated with RMSE and Cramer-von Mises (CM) test (Genest et al., 2009).

Vine copula function is an effective tool for integrating different bivariate distributions and calculating the conditional probability of multiple variables (Ni et al., 2020). In a vine copula, an $n$-dimensional multivariate density is decomposed into

$n(n-1)/2$ bivariate copula densities and arranged into $n$-1 trees. Among numerous vine Copula structures, the C-vine copula





has relatively simple structure and good robustness for constructing multivariate distributions (Wu et al., 2021). Therefore, it was of primary significance to this study. The GOF of C-vine Copulas was estimated with RMSE and CM test. the joint density function of an n-dimensional C-vine Copula is expressed as Equation (10).

$$f\left(x_1,...,x_n\right)=\prod_{m=1}^{n}f_m\left(x_m\right)\times\prod_{i=1}^{n-1}\prod_{j=1}^{n-1}c_{i,i+j|1:(i-1)}\left\{F\left(x_i\mid x_1,...,x_{i-1}\right),F\left(x_{i+j}\mid x_1,...,x_{i-1}\right)\right\}$$

(10)

where $C$ represents cumulative distribution function (CDF) of joint distribution; $F$ represents CDF of marginal distribution. $i$ and $j$ represent root nodes. More detailed information about the n-dimensional C-vine copula can be referred to Wu et al (2021). By this means, the conditional probabilities of ecological drought at different magnitudes under impacts of meteorological drought are calculated using Equation (11).

$$F\left(X>x\mid D>d,A>a,S>s\right)=\frac{F\left(D>d,S>s,A>a,X>x\right)}{F\left(S>s,A>a,D>d\right)}$$

$$=\frac{\begin{array}{l}1-F(d)-F(s)-F(a)-F(x)+C\left(F_D(d),F_S(s)\right)+C\left(F_D(d),F_A(a)\right)\\+C\left(F_D(d),F_X(x)\right)+C\left(F_A(a),F_S(s)\right)+C\left(F_A(a),F_X(x)\right)\\+C\left(F_X(x),F_S(s)\right)-C\left(F_D(d),F_S(s),F_A(a)\right)-C\left(F_D(d),F_S(s),F_X(x)\right)\\-C\left(F_D(d),F_A(s),F_X(x)\right)-C\left(F_S(s),F_A(a),F_X(x)\right)\\+C\left(F_D(d),F_S(s),F_A(a),F_X(x)\right)\end{array}}{\begin{array}{l}1-F_D(d)-F_A(a)-F_S(s)+C\left(F_D(d),F_A(a)\right)+C\left(F_D(d),F_S(s)\right)+\\C\left(F_A(a),F_S(s)\right)-C\left(F_D(d),F_A(a),F_S(s)\right)\end{array}}$$

(11)

where $D$, $A$, and $S$ represent duration, area, and severity of propagated meteorological drought, respectively; $X$ represents ecological drought at moderate, serious, and extreme magnitudes, which equals cumulative probability of 0.5, 0.7, and 0.9, respectively. $C$ represents CDF of joint distribution.

## 4 Results

### 4.1 Top ten meteorological and ecological drought events according to DS

A total of 130 meteorological drought events were identified based on SPI-3 from a three-dimensional perspective. The first ten meteorological drought events in terms of severity in NWC during 1982–2020 are shown in Table 1. Meteorological drought events with longer duration exhibited relatively larger affected area, were mainly concentrated between 1982 and 2000. Zou et al (2005) estimated meteorological droughts with the Palmer drought severity index (PDSI) from 1951 to 2003 in China and found that most parts of NWC experienced severe droughts during 1997–2003, which is similar to the results of this study.

As shown in Table 1, three of ten meteorological drought events occurred during this period. Moreover, according to the historical record, Xinjiang and Gansu province experienced severe meteorological drought during 1985–1986 (Zhang et al., 2019b). The three-dimensional identification method could sensitively capture these events. Event No.9 started from southern Gansu in August 1984 and ended in May 1986, and ranked 1st.





A total of 184 ecological drought events during 1982–2020 were identified using the three-dimensional identification method.

Table 2 lists the top ten ecological drought events in terms of severity. The most severe ecological drought event started in July 1986 and originated from central Gansu, which was induced by the persistent meteorological drought during 1985–1986. Compared with the characteristics of meteorological droughts, ecological droughts showed longer duration and smaller affected area. This reveals that a longer recovery time is required for mitigation of ecological droughts.

**4.2 Identifying propagation from meteorological to ecological drought**

A total of 46 paired drought events were successfully matched based on the spatio-temporal connection criterion. As shown in Fig. 3, points representing paired drought events were mainly distributed along a diagonal, illustrating a relatively high consistency between the two types of droughts on the temporal scale. The number of one–to–one, many–to–one, one–to–many, and many–to–many were 8, 8, 4 and 26, accounting for 17.4 %, 17.4 %, 8.7 % and, 56.5 % of the total number of paired drought events, respectively.

Paired drought event No.36, comprising meteorological drought event No.87 and ecological drought event No.127, was taken as an example to show their spatio-temporal continuums (Fig. 4). The affected area of meteorological and ecological drought in each month were extracted to show their temporal variation. Meteorological drought No.87 (Fig. 4a) started two months ahead of ecological drought (Fig. 4b), and its effects lasted for two months. It is noteworthy that the most severe meteorological and ecological droughts mainly occurred in central Xinjiang. The affected area and severity of meteorological drought event

No.87 and ecological drought event No.127 maintained a similar trend of increase–decrease (Fig. 5). Among them, the peaks of the meteorological drought event appeared two months ahead (December 2007) that of ecological drought (February 2007). In terms of drought trajectory (Fig. 6), they all originated from the Yili Basins and showed a counter clockwise shift.

**4.3 Propagation probability from meteorological to ecological drought**

To estimate the propagated potential of meteorological drought, 11 commonly used machine learning models were trained

based on characteristics of 81 integrated meteorological drought events. Table 3 lists the evaluation results of five-fold cross-validations, including accuracy, precision, recall, F1 Score, and MCC metrics of five-fold cross-validations. The closer these values are to 1, the higher precision of the model. Therefore, the five metrics were summed to compare the performances of the 11 models. Most models showed good performance except for GP and MP. The QDA classifier with maximum total value was chosen as the best model to identify propagation potential of meteorological drought.

The reliability of copula function is highly dependent on the dependence between two variables, which was measured by Kendall's $\tau$ and Spearman's $\rho$ (Chang et al., 2016; Feng et al., 2021). The $\tau$ and $\rho$ between affected area (M_Area), duration (M_Duration), severity (M_Severity) of meteorological drought, and severity of ecological drought (E_Severity) both reached significance at 0.01 level (Table S.1–S.2). The optimal marginal distributions of M_Area, M_Duration, M_Severity, and E_Severity are listed in Table 4. All the distributions passed the KS test and their RMSE were small. Similarly, the parameters

of bivariate distribution were estimated using itau method, and CM test and RMSE were used to evaluate their goodness of fit





(Table 5). The selected bivariate copulas also demonstrated a well applicability. In the end, the C-vine copula was constructed centered on E_Severity. The CM test, RMSE (Table 6), and P–P plots (Fig. S.1) indicated that the distribution can be used in probability analysis. The copula structure of M_Area-M_Duration-M_Severity-E_Severity was shown in Table 6.

Conditional probability is helpful in providing valuable information for effective allocation of water resources under a certain

drought level (Guo et al., 2020). In the current study, the occurrence probabilities of ecological drought at different levels were determined according to the characteristics of meteorological drought (Fig. 7). For example, the occurrence probabilities of moderate, serious, and extreme ecological drought events were 80%, 63%, 14.7%, respectively, when M_DA > $17.6 \times 10^5$ km$^2$ ∩ M_DD > 11.8 month ∩ M_DS > $7.5 \times 10^6$ month·km$^2$. Furthermore, the occurrence probability was found to increase more rapidly with increasing M_DS and M_DD compared with M_DA, indicating that the duration and severity of meteorological

drought had stronger effects on ecological drought than affected area. Additionally, meteorological drought events with a duration of two months but great severity has high potential to trigger ecological drought. This may be attributable to water shortage induced by meteorological droughts with extremely high intensity (intensity is the drought severity divided by the product of drought duration and affected area).

## 5 Discussion

### 5.1 Threshold selection

Determining overlapping areas of drought patches between two adjacent months is critical in identification of drought events from the three-dimensional perspective. Sheffield et al (2009) used 500,000 km$^2$ as the area threshold in global scales. For mainland China, 150,000 km$^2$ was used as the area threshold in some studies (Wang et al., 2011; Xu et al., 2015b). Liu et al (2019) took 1.5% of the total area as the threshold in the Loess Plateau. To determine an optimal area threshold, the number

of meteorological and ecological drought events as well as the ratio of minor drought events were calculated under different area thresholds, respectively. Here, minor drought is defined as a drought event with 2 months duration and average SPI/SEWDI larger than -1. As shown in Fig. 8, 1328 and 2305 meteorological and ecological drought events were identified with an area threshold of 0.48% of the total area of the NWC, and the proportions of minor drought events were 44% and 32%, respectively. The number of drought events and the proportion of minor drought events decreased with increasing area

threshold. However, this trend gradually stabilized when the area threshold was set to be larger than 1.6% of the total area of the NWC, indicating that most minor drought events with relatively small area were excluded. Therefore, 1.6% of the total area of the NWC was used as the area threshold in this study.

Similarly, the sensitivity of $b$ in Eq.(4) for matching two drought types was tested. The binding mode of absolute and relative thresholds was employed to extract spatial intersection. $b$ is set as 10%, 15%, 30%, 50%, 70%, and 90% to match two drought

types. Although some of the successful matching drought events may be merged into one drought event under larger $b$, the number of successful matching drought events showed little difference under different $b$ (Table 7). In the current study, $b$ =15% was set because the most paired drought events could be identified for fitting machine learning models and C-vine copula.





## 5.2 Advantages of proposed approach

Many studies have linked meteorological drought to hydrological drought at different time scales (Ding et al., 2021; Fang et
al., 2020; Feng and Su, 2020; Han et al., 2019; Huang et al., 2017; Ma et al., 2019). In these studies, propagated drought events
were identified on the basis of the time series between two drought types, and they focused on their lagging, attenuation,
lengthening and pooling (Fig. 9a). Spatial and temporal drought propagation identification method used in the current study
not only preserved the characteristics identified by low dimensional method, but also considered the spatial overlap of two
drought types (Fig. 9b). Using this method, two types of drought events without spatial connection would be excluded, and
more drought characteristics, such affected area and migration path could be extracted. This addresses the limited applicability
of the traditional method to regions with large spatial extent, and provides more realistic drought propagation information.
Additionally, we improved the method for calculating affected area and duration of paired drought events developed by Liu et
al (2019), represented by a simple sum of characteristics of multiple drought events. However, this method overestimates the
duration and affected area of some paired drought events, which is inconsistent with the real situation. In this study, the
enhanced method could reflect the characteristics of paired drought during the propagation process more accurately.
The conditional probability model was constructed based on paired meteorological and ecological drought events; it is not
suitable for calculating the probability of ecological drought at different levels according to meteorological drought events
without propagation potential. For example, the probability of moderate ecological drought was 63.3% if the characteristics of
meteorological drought event No.122 (M_Area=$5.1\times10^5$ km$^2$, M_Duration=6 month, M_Severity=$1.89\times10^6$ month·km$^2$) was
directly input to the conditional probability model. In reality, this meteorological drought event did not trigger ecological
drought. The QDA model added before the C-vine copula was used to address this issue, which could estimate the propagation
potential of the corresponding meteorological drought. After this modification, the probability of the propagation of
meteorological drought event No.122 to moderate ecological drought changed to 24.8%.

## 5.3 Uncertainty of the model and its improvement measures

QDA model could well simulate the propagation potential of most meteorological drought events (Table 3). However, some
errors occurred in humid southern Shaanxi. For example, meteorological drought events No. 22 showed the potential to trigger
ecological droughts, which were incorrectly classified as propagation occurrence. This could be attributed to the compensation
of rich water resources for short-term ecological water deficit. Additionally, this paper provides a method for estimating the
occurrence probability of ecological drought under the condition of a certain precipitation deficit. The effects of human
activities and climate change on ecological drought were not distinguished in the current study. The proposed method may not
be accurate for regions with complex water supply systems and strong anthropogenic impacts on vegetation growth.
To improve the accuracy of the method, future studies should consider the non-consistence of ecological drought to quantify
the impacts of human activities on drought propagation. Moreover, SPI can be replaced by PDSI or scPDSI to represent
meteorological drought through which multiple water balance processes are considered to analyze their relationship with



ecological drought (Altunkaynak and Jalilzadnezamabad, 2021). However, such modification may lead new problem associated with spatio-temporal incomparability. Nevertheless, this approach is worth applying in the ecological drought warning. For example, when a meteorological drought event occurs, its characteristics can be applied as input to the tuned model to estimate propagation probability from meteorological to ecological drought in different degrees.

## 6. Conclusions

This study proposed a method in identifying the propagation probability of meteorological drought events to trigger ecological drought in different magnitudes. Taking NWC as an example, 130 meteorological drought and 185 ecological drought events during 1982–2020 were extracted using the three-dimensional identification method. Compared with meteorological drought, ecological drought events exhibited longer duration, but smaller affected area and severity, suggesting that a longer recovery time is required for mitigating ecological drought.

A total of 46 drought events were successfully matched according to a certain spatio-temporal connection principle. The paired drought events were divided into four categories, including one-to-one, many-to-one, one-to-many, and many-to-many. The four categories accounted for 17.4 %, 17.4 %, 8.7 %, and 56.5 % of the total number of paired drought events, respectively. Then, a drought propagation probability model was constructed by coupling QDA and C-vine copula. Compared with the traditional propagation probability model, the proposed model intuitively provides more objective probabilities of ecological

drought at different magnitudes.

The current study certainly provides a more robust method for estimating propagation probability from meteorological to ecological drought in similar ecologically fragile regions.

## Data availability

Monthly meteorological data, including surface reflectance, temperature, relative humidity, atmospheric pressure, downward

shortwave radiation, wind speed, and longwave radiation, obtained from the ERA5-land reanalysis dataset (https://cds.climate.copernicus.eu) issued by European Centre for Medium-Range Weather Forecasts (ECWMF), which has a spatial resolution of 0.1° × 0.1° and covers the period of 1981–2021. Root soil moisture data were obtained from the hydrological dataset, simulated by Noah model of the Global Land Data Assimilation System (GLDAS, 0.25°×0.25°; https://ldas.gsfc.nasa.gov/gldas), covering the period of 1948–2021. NDVI data covering the period 1981–2021 were obtained

from the National Centers for Environmental Information (NCEI) (https://www.ncei.noaa.gov/), with a spatial resolution of 0.05° × 0.05°. Land use type data (LUTD) with a spatial resolution 1 km was downloaded from China's multi-period land use/cover change monitoring dataset (http://www.resdc.cn); it includes years of 1980, 1990, 1995, 2000, 2005, 2010, 2015, 2018, and 2020.



**Author contribtion**

Tianliang Jiang: Conceptualization, Methodology, Software, Visualization, Writing - original draft. Xiaoling Su: Data curation, Validation, Investigation, Funding acquisition, Supervision, Formal analysis. Gengxi Zhang: Writing - review & editing, Supervision. Te Zhang: Formal analysis, Investigation. Haijiang Wu: Data curation, Investigation.

**Competing interests**

The authors declare that they have no conflict of interest.

**Acknowledgement**

The study was supported by the National Natural Science Foundation in China (Grants 52079111 and 51879222).

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



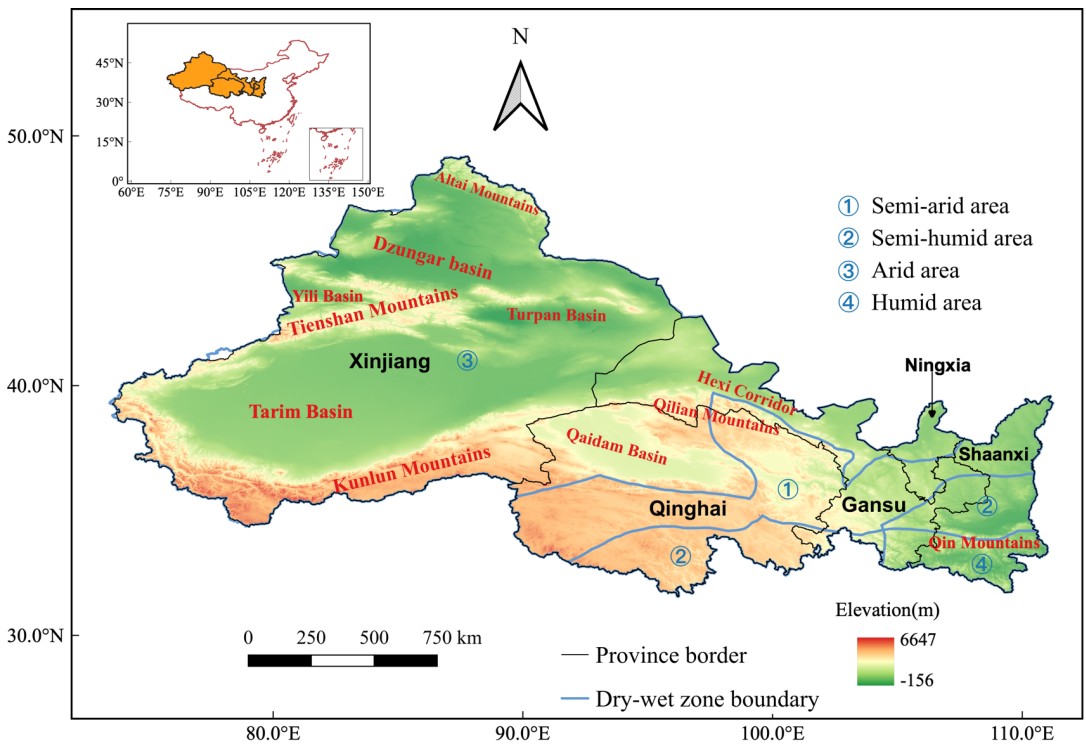

**Figure 1: Elevation and four divisions of northwestern China.**





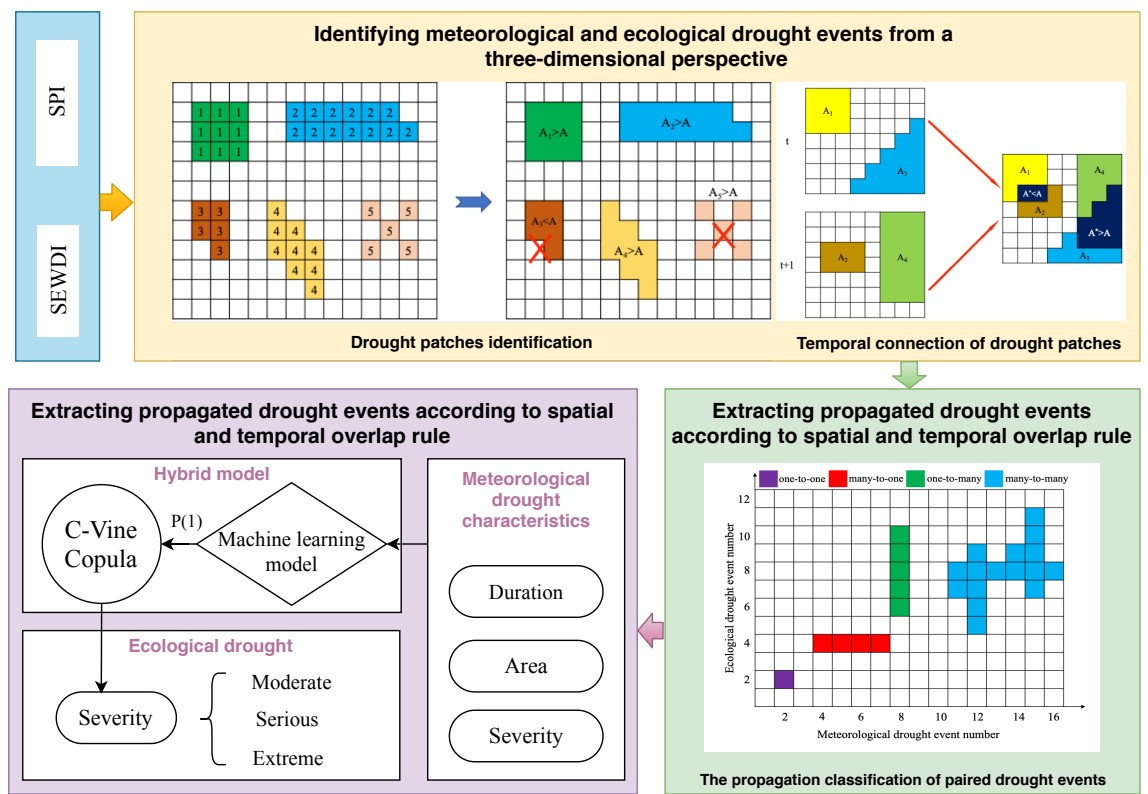


**Figure 2: A schematic diagram illustrating the procedure of the drought propagation identification method.**

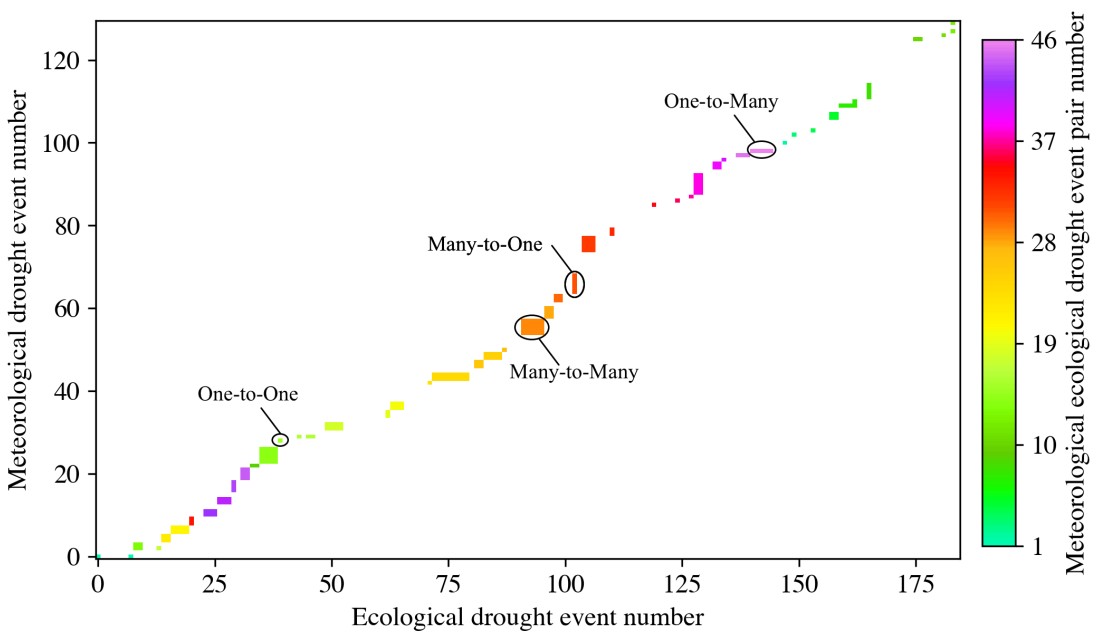





**Figure 3: Identification results of paired meteorological and ecological drought events.**

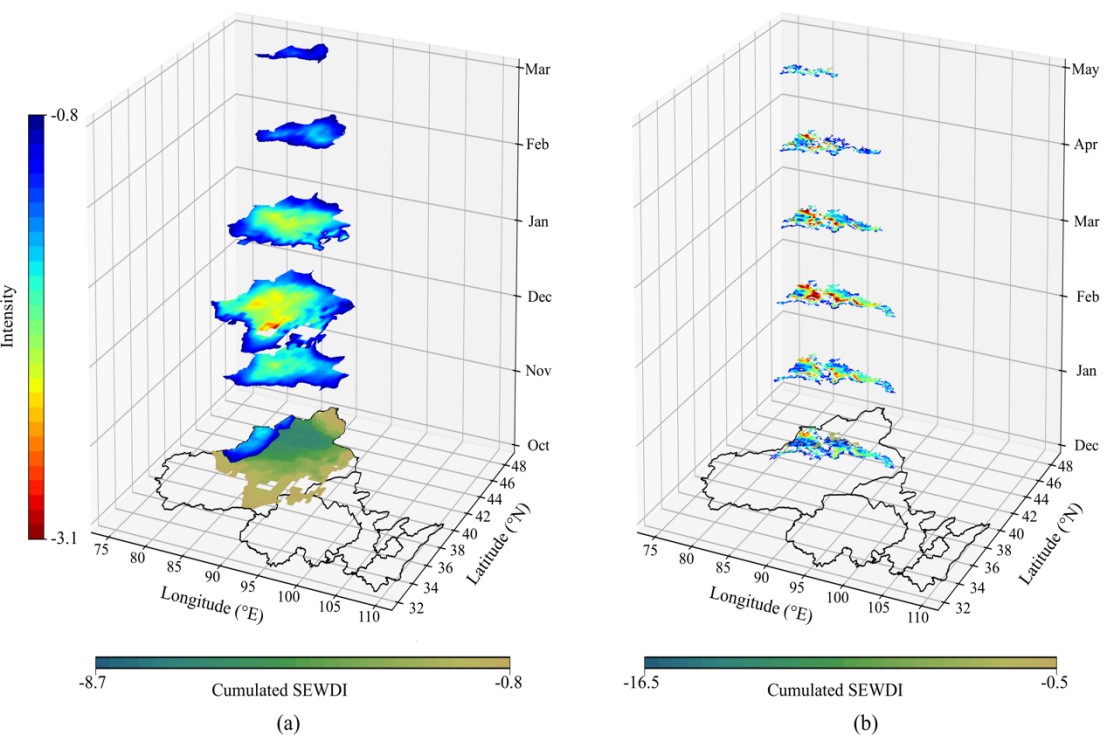

(a)                                                         (b)

**Figure 4: Spatio-temporal continuums of (a) meteorological drought event No. 87 and (b) ecological drought event No. 127.**

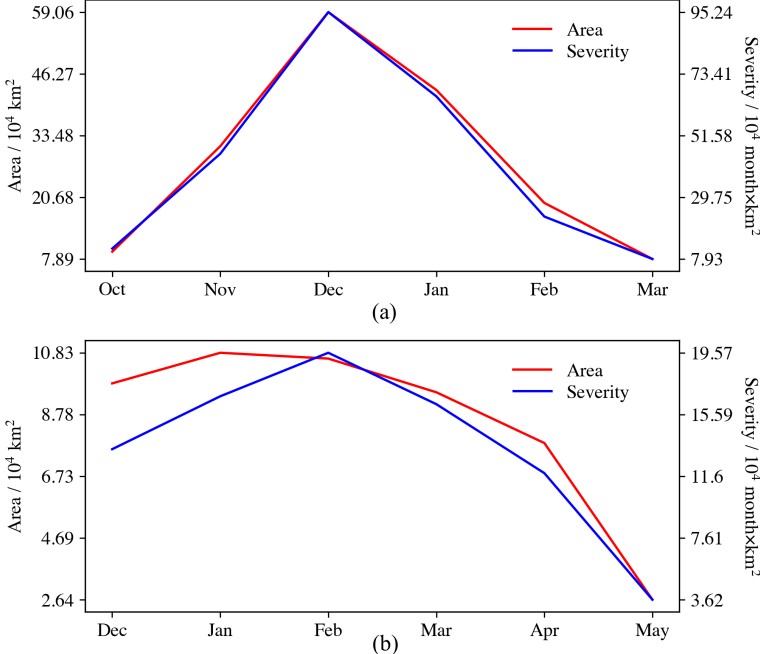

**Figure 5: Temporal evolution of DS and DA of (a) meteorological drought event No. 87 and (b) ecological drought event No. 127.**



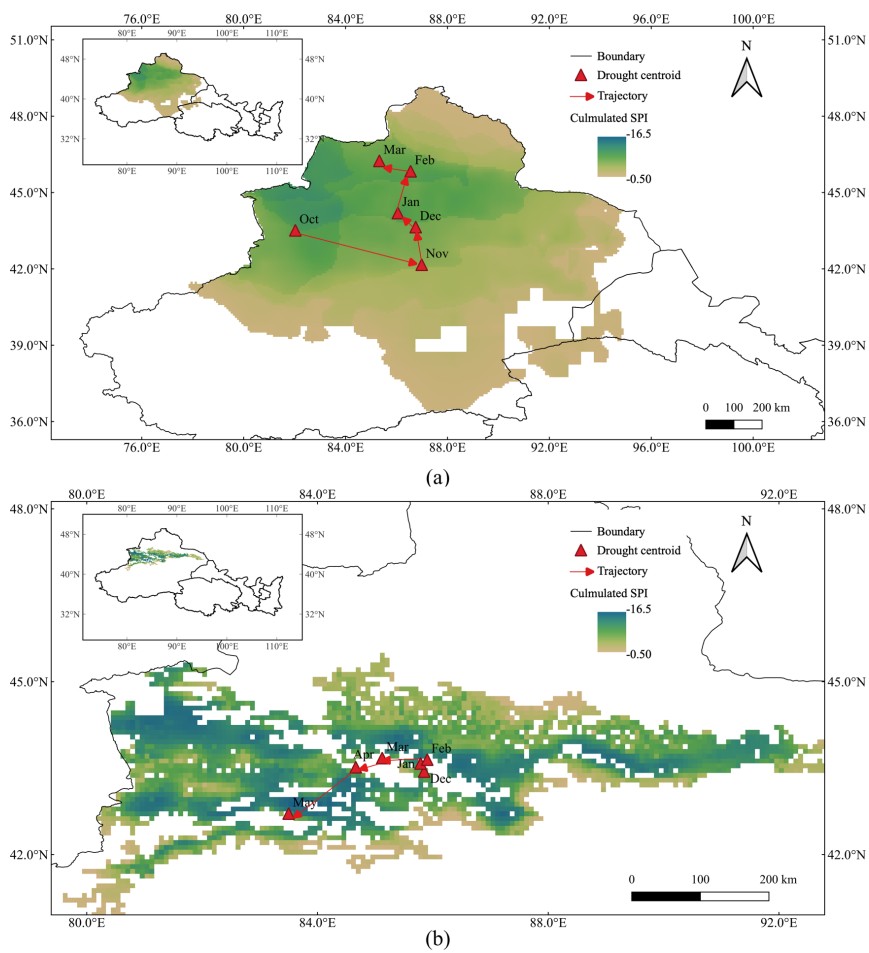

**Figure 6: Cumulative SPI/SEWDI and migration trajectory of (a) meteorological drought event No.87 and (b) ecological drought**
**event No.127.**





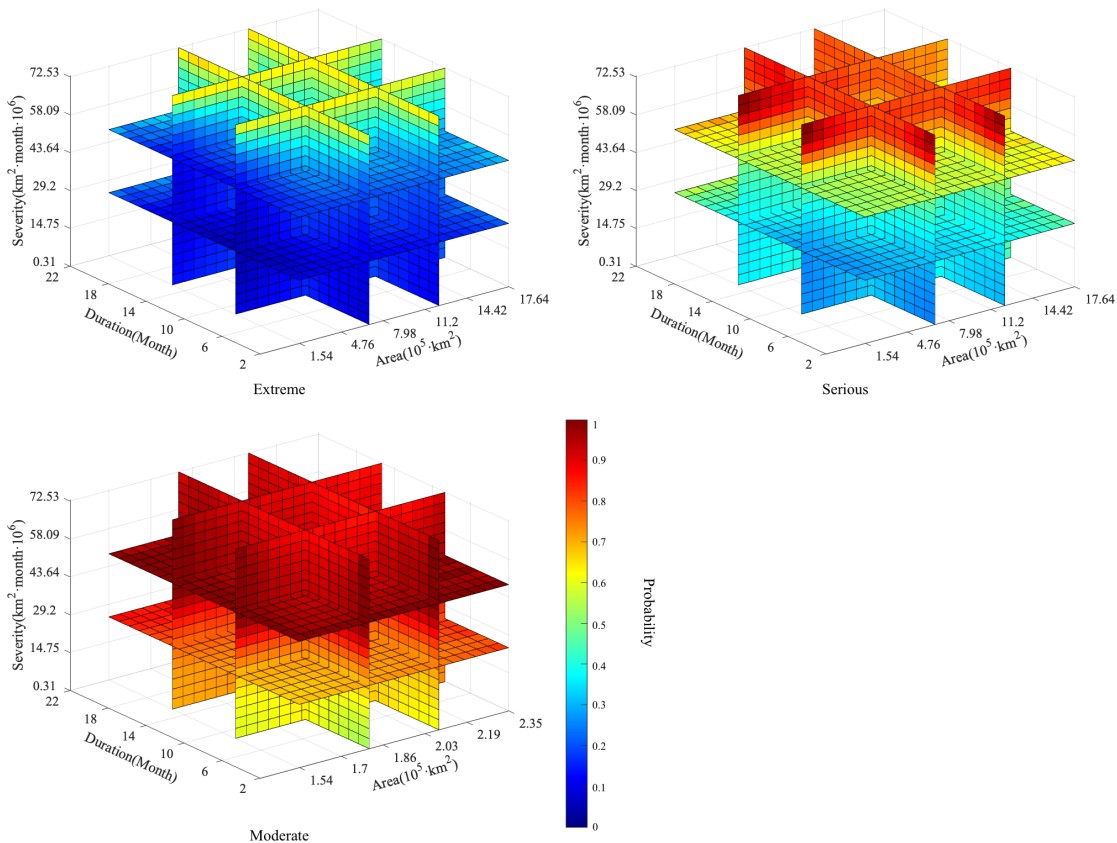

**Figure 7: Conditional probability of ecological drought at different levels given that characteristics of meteorological drought exceeding a certain value.**





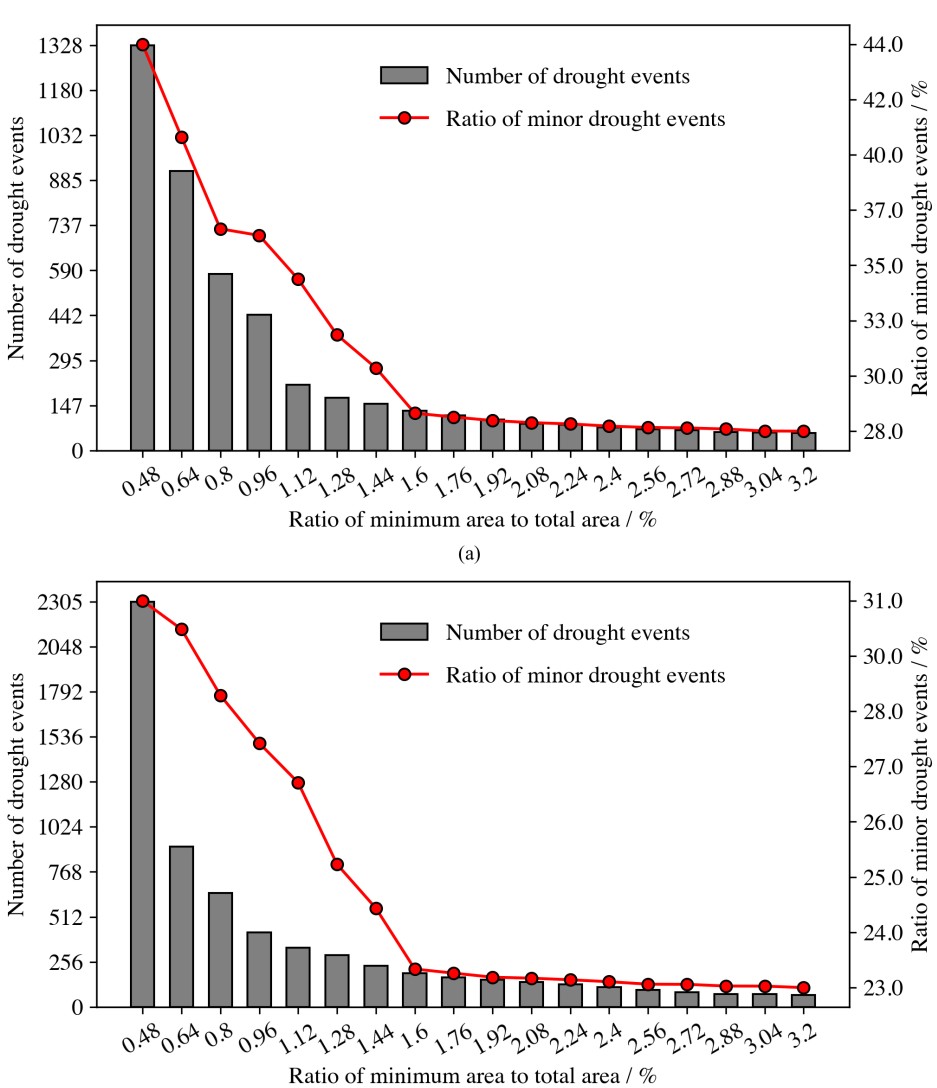


**Figure 8: Sensitivity test of overlapping areas of drought patches between two adjacent months.**





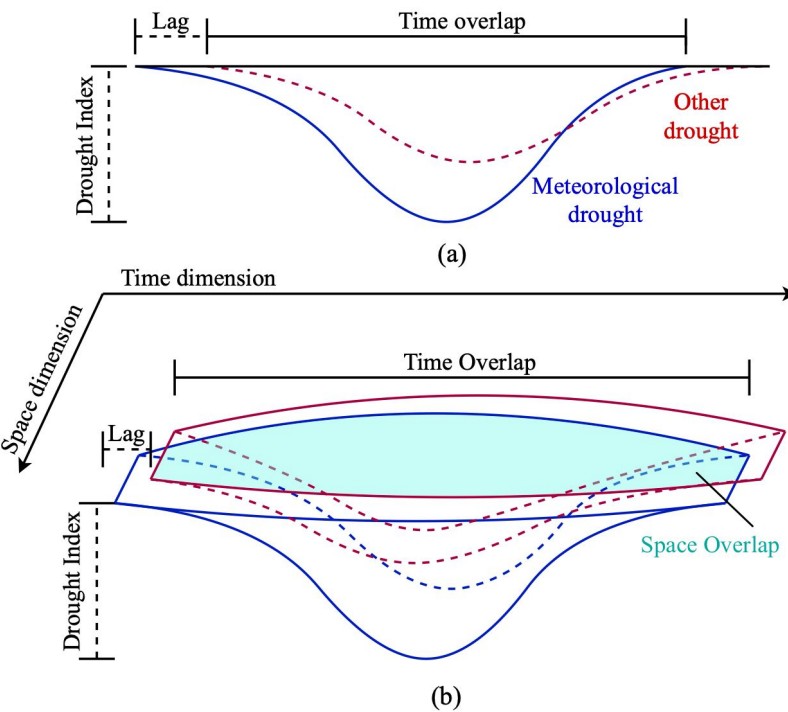

**Figure 9: Conceptual graph depicting (a) traditional and (b) spatial and temporal connectivity rule of two drought types.**

**Table 1 Top ten meteorological drought events according to severity**

| No. | Affected area (km²) | Duration (month) | Severity (month · km²) | Start time (year–month) | End time (year–month) |
|---|---|---|---|---|---|
| **22** | 1764139.2 | 22 | 70730086.95 | 1985–08 | 1987–05 |
| **15** | 1675168.85 | 12 | 35741813.93 | 1997–01 | 1997–12 |
| **74** | 1511945.05 | 8 | 29594926.83 | 2001–02 | 2001–09 |
| **88** | 1610083.95 | 6 | 28099641.92 | 2008–04 | 2008–09 |
| **54** | 1613507.4 | 8 | 21190554.23 | 1983–10 | 1984–05 |
| **46** | 1407943 | 6 | 19184637.61 | 1995–03 | 1995–08 |
| **23** | 1553813.45 | 6 | 17818477.77 | 1986–07 | 1986–12 |
| **64** | 1194351.2 | 4 | 17346922.36 | 2000–02 | 2000–05 |
| **120** | 954552.3 | 8 | 2642278.555 | 2017–10 | 2018–05 |
| **115** | 471019.5 | 5 | 1915975.753 | 2015–12 | 2016–04 |

**Table 2 Top ten ecological drought events according to the severity**



| No. | Affected area (km²) | Duration (month) | Severity (month · km²) | Start (year–month) | End (year–month) |
|---|---|---|---|---|---|
| **35** | 351159.4 | 24 | 12326328.57 | 1986–07 | 1988–06 |
| **49** | 407902.1 | 30 | 10498396.17 | 1990–06 | 1992–01 |
| **72** | 156298.2 | 8 | 5731890.703 | 1994–07 | 1995–02 |
| **91** | 435801.25 | 20 | 4671546.994 | 1997–02 | 1998–10 |
| **2** | 390824.2 | 25 | 4461612.098 | 1982–04 | 1984–04 |
| **59** | 407626.65 | 21 | 3562570.147 | 1991–03 | 1992–01 |
| **81** | 371582.05 | 8 | 3309776.519 | 1995–03 | 1995–10 |
| **96** | 364656.45 | 13 | 3275325.857 | 1997–07 | 1998–07 |
| **4** | 358006.3 | 10 | 2465549.482 | 1982–04 | 1983–01 |
| **83** | 409436.75 | 10 | 2377022.171 | 1995–04 | 1996–01 |

**Table 3 Estimations of 11 machine learning models in identifying the potential of meteorological drought to trigger ecological drought**

| Classifier | Accuracy | Precision | Recall | F1 Score | MCC | Total |
|---|---|---|---|---|---|---|
| KN | 0.89 | 0.89 | 0.91 | 0.89 | 0.80 | 4.38 |
| SVM | 0.80 | 0.84 | 0.83 | 0.80 | 0.67 | 3.94 |
| GP | 0.43 | 0.22 | 0.50 | 0.30 | 0.00 | 1.45 |
| DT | 0.83 | 0.84 | 0.84 | 0.82 | 0.68 | 4.02 |
| MP | 0.62 | 0.40 | 0.59 | 0.46 | 0.17 | 2.24 |
| AB | 0.82 | 0.83 | 0.82 | 0.81 | 0.65 | 3.92 |
| GNB | 0.85 | 0.88 | 0.88 | 0.85 | 0.75 | 4.21 |
| **QDA** | **0.93** | **0.93** | **0.94** | **0.93** | **0.87** | **4.58** |
| GB | 0.83 | 0.83 | 0.84 | 0.82 | 0.67 | 4.00 |
| XGB | 0.85 | 0.86 | 0.87 | 0.85 | 0.72 | 4.15 |
| RF | 0.87 | 0.87 | 0.89 | 0.86 | 0.76 | 4.25 |

**Table 4 Goodness of fit of the marginal distribution**

| Distribution | Marginal distribution | RMSE | KS-test | |
|---|---|---|---|---|
| | | | Statistics | P-value |
| M_Area | Johnsonsb | 0.044 | 0.129 | 0.963 |
| M_Duration | Johnsonsb | 0.068 | 0.161 | 0.823 |
| M_Severity | Pearson III | 0.057 | 0.226 | 0.413 |
| E_Severity | Johnsonsb | 0.079 | 0.194 | 0.615 |






**Table 5 Goodness of fit of the bivariate distribution**

| | Copula | RMSE | CM_test | |
|---|---|---|---|---|
| | | | Statistic | P-value |
| M_Area–M_Duration | Frank | 0.005 | 0.086 | 0.373 |
| M_Area–M_Severity | Gaussian | 0.052 | 0.098 | 0.605 |
| M_Area–E_Severtiy | Gumbel | 0.032 | 0.042 | 0.933 |
| M_Duration–M_Severity | Gaussian | 0.057 | 0.102 | 0.585 |
| M_Duration–E_Severity | Gaussian | 0.053 | 0.087 | 0.663 |
| M_Severity–E_Severity | Frank | 0.054 | 0.105 | 0.570 |

**Table 6 Goodness of fit of the multivariate distribution**

| | RMSE | CM_test | |
|---|---|---|---|
| | | Statistic | P-value |
| M_Area–M_Duration–M_Severity–E_Severity | 0.079 | 0.073 | 0.398 |

**Table 7 Sensitivity test of parameter *b***

| Threshold | Number of paired drought events |
|---|---|
| min($AM_i$,$AH_j$)*90% | 23 |
| min($AM_i$,$AH_j$)*70% | 32 |
| min($AM_i$,$AH_j$)*50% | 36 |
| min($AM_i$,$AH_j$)*30% | 39 |
| min($AM_i$,$AH_j$)*15% | 46 |
| min($AM_i$,$AH_j$)*10% | 46 |