# Peer review of "Estimating propagation probability from meteorological to ecological droughts using a hybrid machine learning-Copula method"

_Hydrology and Earth System Sciences, 2022_

## Author Comment (AC1)

Dear Referee #1:

Thank you for your kind and encouraging comments on our study. Your comments and our responses to them are listed below.

- Powerful machine learning approaches were applied. What were the degrees of freedom of the machine learning approaches? What is the ratio of the degrees of freedom over the rather small number of 81 meteorological drought events?

RESPONSE: It seems these two problems were posed in order to investigate whether more degrees of freedom would cause the model to overfit the training data. In general, Regularization techniques and optimal model architectures are employed to ensure machine learning models are not overfitted and maintain low generalization errors. Therefore, degrees of freedom and model complexity always correspond very poorly (Janson et al., 2013), which is generally much less than the number of parameters in the model (Gao and Jojic, 2016). In this study, we used a Python package called PyCaret to construct these classifiers. L2 regularization method was selected in each model to avoid overfitting and maintain high calculation efficiency. The relevant description will be added to Section 2.4.3 in the next version.

MODIFICATION: In this study, each binary classifier was constructed using a Python package called PyCaret, which wraps several machine-learning libraries, including scikit-learn, XGBoost, LightGBM, CatBoost, spaCy, Optuna, and Hyperopt(Ali, 2020). It is simple to select the optimal hyperparameters of each model using the tune_model() function in PyCaret package. A 5-fold cross-validation was used to train and validate the classifiers in each model by setting "fold=5" in create_model(). In using compare_models() function, the classifier with the highest summation of accuracy, precision, recall, F1 score, and Matthews correlation coefficient was selected as the optimal model. To avoid overfitting and maintain high calculation efficiency, L2 regularization method was selected in each model by setting parameter "penalty='l2'".

- According to Fig. 7 propagation probability is nearly exclusively determined by the

severity of the meteorological drought which would meet common expectations. In contrast, any effect of duration or area is hardly discernible. Please compare the performance of the machine learning approaches to that of a multivariate linear regression

RESPONSE: We agree and can see your point. In this study, machine learning models were used to determine whether a meteorological drought event has propagating potential. It is therefore a binary classification question. We will add Figure 9 and a description related to it in Section 3.4 in the next version,.

MODIFICATION:

[Figure]

Figure 9 Three-dimensional diagram showing characteristics of meteorological drought events. Larger circles indicate greater severity.

As can be seen in Figure 9, propagated meteorological droughts have greater severity, larger affected area, and longer duration than non-propagated droughts.

RESPONSE: We agree that any effect of duration or area is hardly discernible. In this study, meteorological drought and ecological drought with genetic relationship were extracted on the basis of a certain spatio-temporal matching rule. Therefore, the model constructed in this study only includes meteorological drought and ecological drought events that have genetic relationships. As a result, only 103 out of 184 ecological drought events were

induced by 81 out of 108 meteorological drought events. Severity of ecological drought thus can be predicted based on the characteristics of meteorological drought. We will add relevant description below in Section 4.1 in the next version.

MODIFICATION: Using this method, two types of drought events without spatial connection would be excluded (only 103 out of 184 ecological drought events were induced by 81 out of 108 meteorological drought events), and more drought characteristics, such as affected area and migration path could be extracted. This addresses the limited applicability of the traditional method to regions with large spatial extent, and provides more reliable information for quantifying relationship between characteristics of meteorological drought and ecological drought.

RESPONSE: We will include your recommendation and add multivariable linear regression in Section 3.4 in the next version.

MODIFICATION: As a comparison, ternary linear model and ternary quadratical model were constructed based on 46 pairs of meteorological-ecological droughts (Table 7). There three independent variables, M_DS, M_DD, and M_DA, and one dependent variable, E_DS. From Table 7, it is evident that the $R^2$ of ternary quadratic model is higher than ternary linear model, and RMSE, AIC, and BIC are lower. This illustrates that M_DS, M_DD, M_DA and E_DS follow a nonlinear relationship, and ternary quadratic model is more suitable for simulating their relationship. Using the ternary quadratic model, E_DS equals $1.4 \times 10^6$ month·km$^2$ when M_DA > $17.6 \times 10^5$ km$^2$ ∩ M_DD > 11.8 month ∩ M_DS > $7.5 \times 10^6$ month·km$^2$. It falls below the thresholds of moderate ($1.7 \times 10^6$ month·km$^2$), severe ($2.4 \times 10^6$ month·km$^2$), and extreme ($4.6 \times 10^6$ month·km$^2$) ecological drought.

**Table 8 Modelling E_DS with polynomial functions based on meteorological drought characteristics**

| Model types | Expression | Assessment metrics | | | |
|---|---|---|---|---|---|
| | | RMSE | AIC | BIC | $R^2$ |
| Ternary linear model | $E\_DS=4.85\times10^5+0.15M\_DS+4099.35M\_DD-1.20M\_DA$ | $9.24\times10^5$ | 1350.67 | 1357.89 | 0.58 |
| Ternary quadratic model | $E\_DS=1.54-0.05M\_DS-16.91M\_DD-0.08M\_DA-1319.23M\_DD^2+0.03M\_DD\times M\_DA$ | $7.29\times10^5$ | 1085.75 | 1100.20 | 0.85 |

● Please check the use of definite and indefinite articles and the use of plural "s".

RESPONSE: Thanks for the hint. We will check them carefully to avoid grammar errors in the next version.

Details:

● 53-55: Who is "they"?

RESPONSE: Thanks for the hint. The sentence "In other words, they considered temporal connection of two drought types and ignored their spatial overlap, which may result in the miscalculation of drought propagation in regions with large spatial extent." has been changed to "In other words, traditional statistical method only considered temporal connection of two drought types and ignored their spatial overlap, which may result in the miscalculation of drought propagation in regions with large spatial extent."

● 73-79: Section "2 Study area" comprises only 6 lines and should be merged with the subsequent section 3, or at least with section "3.1 Datasets".

RESPONSE: We agree. Section 2 will become Section 3.1 in Section 3 in next version.

● 82-85: Verb is missing.

RESPONSE: Thanks for the hint. The sentence " Monthly meteorological data, including surface reflectance, temperature, relative humidity, atmospheric pressure, downward shortwave radiation, wind speed, and longwave radiation, obtained from the ERA5-land reanalysis dataset (https://cds.climate.copernicus.eu) issued by European Centre for Medium-Range Weather Forecasts (ECWMF), which has a spatial resolution of $0.1^\circ \times 0.1^\circ$

and covers the period of 1981–2021" has been changed to " Monthly meteorological data, including surface reflectance, temperature, relative humidity, atmospheric pressure, downward shortwave radiation, wind speed, and longwave radiation, was obtained from the ERA5-land reanalysis dataset (https://cds.climate.copernicus.eu) issued by European Centre for Medium-Range Weather Forecasts (ECWMF), which has a spatial resolution of 0.1° × 0.1° and covers the period of 1981–2021."

- 91: Use lowercase letter in "Root".

RESPONSE: Thanks for the hint. We have corrected it to "root".

- 98: Replace "deep phreatic buried depth" by "great depth to groundwater".

RESPONSE: Thanks for the hint. We have corrected it to " great depth to groundwater ".

- 112: Both "SEWDI" and "SEBS" need to be explained in a concise way. Referring to the Jiang et al. (2021) paper does not suffice.

RESPONSE: We agree and can see your suggestion. We will explain it in next version.

MODIFICATION: SEWDI follows a similar procedure as SPI, which includes the calculation of Ecological water deficit (EWD), the selection of an optimal distribution for fitting monthly EWD series, and the inverse normal transformation of cumulative density distribution of EWD. EWD is the difference between ecological water requirement (EWR) and ecological water consumption (EWC) (Chi et al., 2018; Jiang et al., 2021). Among them, the EWR was calculated using the single crop coefficient method recommended by the Food and Agriculture Organization (FAO). EWC equals the actual evapotranspiration, which is derived from latent heat fluxes calculated by the surface energy balance system (SEBS) algorithm.

- 124: Should be "three steps", not "two steps".

RESPONSE: Thanks for the hint. We have corrected it to " three steps ".

- 147: Delete "to".

RESPONSE: Thanks for the hint. We have deleted "to".

- 200: Do you mean "Johnson S_B distribution"?

RESPONSE: Thanks for the hint. We have corrected "johnsonsb" to "Johnson S_B" in the full text.

- 224: What does "DS" mean?

RESPONSE: DS represent drought severity. We would change the name to its full form.

- 265: Please explain "itau method".

RESPONSE: We have added "The itau method makes parameter estimation for Copula easier by inverting Kendall's tau method (Demarta and McNeil, 2005)." for explaining "itau method".

- 280-297: Section 5.1 should be either part of the methods or of the results section.

RESPONSE: Thanks for your suggestion, we will move Section 5.1 to the results section (Section 3.1) in next version.

- 349-352: Verb is missing.

RESPONSE: Thanks for the hint. The verb has been added to this sentence.

MODIFICATION: Monthly meteorological data, including surface reflectance, temperature, relative humidity, atmospheric pressure, downward shortwave radiation, wind speed, and longwave radiation, was obtained from the ERA5-land reanalysis dataset (https://cds.climate.copernicus.eu) issued by European Centre for Medium-Range Weather Forecasts (ECWMF), which has a spatial resolution of $0.1° \times 0.1°$ and covers the period of 1981–2021.

- Figure 3: I guess that the drought event numbers reflect chronological order, is that right? The colour scale indicates about the same meteorological-ecological drought event number for very different ecological and meteorological drought event numbers. E.g., green symbols show up for ecological drought event number 1-10, 30-50 and >150. How can that be? Is there something wrong with the colour coding of the symbols?

RESPONSE: You are right, we have corrected this mistake. Figure 3 will be replaced with the figure below in the next version.

[Figure]

Figure 4: Identification results of paired meteorological and ecological drought events

- Figure 7: In the figure caption correct "exceeding" to "exceed".

RESPONSE: Thanks for the hint. We have corrected "exceeding" to "exceed".

References:

Ali, M.: PyCaret, PyCaret: An open source, low-code machine learning library in Python, 2020.

Demarta, S. and McNeil, A. J.: The t Copula and Related Copulas, International Statistical Review / Revue Internationale de Statistique, 73, 111–129, 2005.

Gao, T. and Jojic, V.: Degrees of Freedom in Deep Neural Networks, https://doi.org/10.48550/arXiv.1603.09260, 2016.

Janson, L., Fithian, W., and Hastie, T.: Effective Degrees of Freedom: A Flawed Metaphor, https://doi.org/10.1093/biomet/asv019, 2013.

---

## Author Comment (AC2)

Dear Referee #2:

First of all, thank you very much for the helpful comments and your kind feedback, your comments are listed below, along with our responses to each one.

- Some works, e.g., the systematically literature review of propagation probability model, need to be added into the Introduction or Discussion sections.

RESPONSE: Thanks for your advice. The purpose of this section is to introduce the use of Bayesian model to explore probabilistic relationships among different types. We have rewritten this section based on your advice.

MODIFICATION: A number of studies have attempted to assess the propagation relationships between two drought types based on probabilistic method. This is due to that the probability information of one type of successive drought events is contained in another type of drought associated with it (Wu et al., 2021). Bayesian networks is a probabilistic model that acquires probabilistic inferences over interacting variables of interest based on a graphical structure. Therefore, this method has been proved to be a better way to quantify the probability relationship between different drought types (Ayantobo et al., 2018; Chang et al., 2016; Das et al., 2020). For example, Guo et al (2020) calculated occurrence probability of hydrological drought based on different intervals of duration and severity of meteorological drought. Sattar et al (2019) identified the occurrence probability of different classes and lag time of hydrological drought according to intensity of meteorological drought. Xu et al., (2021) found that the probability of agricultural drought severity increased synchronously with meteorological drought in different regions of China. Jehanzaib et al., (2020) concluded that the probability of meteorological drought in the Korean Peninsula propagating into hydrological drought increased significantly under climate change. In general, these studies primarily focus on the relationship between duration and severity between two drought types but ignore their affected area relationships. Xu et al. (2015a) found that drought occurrence probability would be underestimated if drought affected area were not considered. Therefore, the traditional

drought probabilistic model of drought propagation can be improved by introducing the three-dimensional drought identification method, which provides more drought information (Liu et al., 2019).

- Line 214 (i.e., Eq. (10)), the "n – 1" is shown in the inner product should be revised as "n – i". Moreover, please define or explain the symbols that appeared in this and some other equations, e.g., the term c is not defined. Please carefully check them.

RESPONSE: Thanks for your hint. We indeed made a mistake when typing the Eq.(10). We have corrected it and added the explanation of the symbols. Furthermore, all equations have been carefully checked to ensure no errors would occur.

MODIFICATION:

$$f(x_1,\ldots,x_n) = \prod_{i=1}^{n} f_i(x_i) \times \prod_{i=1}^{n-1}\prod_{j=1}^{n-i} c_{i,i+j|1:(i-1)}\left\{F(x_i \mid x_1,\ldots,x_{i-1}), F(x_{i+j} \mid x_1,\ldots,x_{i-1})\right\} \tag{10}$$

where $f(x_1,...,x_n)$ represents the joint density function. $c$ represents bivariate Copula densities, which includes Gumbel, Gaussian, Frank, and Clayton Copula function; $F$ represents cumulative distribution function of marginal distribution. $i$ and $j$ represent root nodes.

- For Section 4.2, please provide specific information about paired drought events so that we can identify the characteristics of four paired categories. At the same time, I found the statement "Among them, the peaks of the meteorological drought event appeared two months ahead (December 2007) that of ecological drought (February 2007)" may be wrong in Lines 250-251. As you know, the duration between December 2007 and February 2007 is far more than two months. Please revise it.

RESPONSE: This is a good suggestion. We have provided specific information about paired drought events to show the characteristics of four paired categories. Relevant description will be added in Section 3.3 in the next version.

MODIFICATION:    In type OTM, meteorological drought showed a longer duration, a larger affected area, and a greater severity than ecological drought. However, this is contrary to

the type of MTO. Simultaneously, ecological drought in type MTO showed a longer duration, a larger affected area, and a greater severity than those in type OTM.

[Figure]

Figure 5. A box plot showing the intensity, duration, and affected area of paired meteorological-ecological drought among different types

MODIFICATION: The statement "Among them, the peaks of the meteorological drought event appeared two months ahead (December 2007) that of ecological drought (February 2007)" was corrected to " Among them, the peaks of the meteorological drought event appeared two months ahead (December 2007) that of ecological drought (February 2008)."

● An essential part of this article is the use of machine learning to solve a binary

classification problem. In this context, I suggest adding a plot that shows the severity, duration, and affected area of meteorological droughts propagated and didn't propagate.

RESPONSE: That is an excellent suggestion. Thank you very much for that as well! In this study, machine learning models were used to determine whether a meteorological drought event has propagating potential. We have added Figure 7 and related description in Section 4.3 with reference to the advice of Referee#1.

MODIFICATION:

[Figure]

Figure 9 Three-dimensional diagram showing characteristics of meteorological drought events. Larger circles indicate greater severity.

As can be seen in Figure 9, propagated meteorological droughts have greater severity, larger affected area, and longer duration than non-propagated droughts.

- I suggest replacing "Three-dimensional drought identification method" with "Three-dimensional clustering method".

RESPONSE: We agree and can see your point. We will replace "Three-dimensional drought identification method" with "Three-dimensional clustering method" in the next version.

Details:

- Lines 24-25, for specification, the drought classification should be changed as the "drought types". Please revise it.

RESPONSE: Thanks for the hint. We have corrected "drought classification" to "drought types".

- Lines 66-68, "Taking a typically ecological fragile region … to meteorological drought", I suggest revising it as "Taking a typically ecological fragile region, Northwestern China (NWC), as an example, the motivation of this study, from a three-dimensional perspective, is proposed a novel hybrid machine learning-Copula method to investigate the response probability of ecological drought to meteorological drought" to highlight the novelty of this paper.

RESPONSE: Thanks. We have followed your advice, and changed it.

- Line 94-99, authors should refine their explanation of why SPI-3 is used to represent meteorological drought.

RESPONSE: Thanks for the hint. Line 94-99 has been changed to " Previous studies found that the standardized precipitation evaporation index (SPEI) overestimated the meteorological drought in NWC where actual atmospheric water demand is determined by precipitation variation (Ayantobo and Wei, 2019; Zhang et al., 2019a; Zhang et al., 2021b). Additionally, precipitation is the main water resources for vegetation growth in most regions of NWC due to the great depth to groundwater (Cao et al., 2021). Standardized precipitation index (SPI) was thus used in the current study to represent meteorological drought. SPI at different time scales was calculated by aggregating $n$-month moving sums, allowing the identification of various drought types (McKee et al., 1993). At short time scales, drought events have a high frequency and a short duration, while at the longest time scales, they have a longer duration and a lower frequency. SPI–3 has been reported to be highly representative of the impacts of meteorological conditions on vegetation as the vegetation variation is sensitive to three months accumulated precipitation (McKee et

al., 1993; Vicente-Serrano et al., 2012; Vicente-Serrano et al., 2010; Vicenteserrano et al., 2010). Therefore, SPI-3 was used to characterize meteorological drought in this study."

- Section 3.3.2, the number of steps regarding the Spatiotemporal connection of two drought types may be disordered, e.g., the statements of steps were listed as Firstly and Secondly in Lines 154-155, but that remained as the Secondly in Line 163. Please check it.

  RESPONSE: Thanks for the hint. We will revise "Secondly" in line 163 in original manuscript to "Thirdly" in next version.

- Lines 206-207, please revised the "Cramer-von Mises (CM) test" as the "Cramer-von Mises (CvM) test" based on common sense.

  RESPONSE: Thanks for the hint. We have corrected it to " Cramer-von Mises (CvM) test".

- Line 224, I recommended the authors revised the caption of this section as "Top ten meteorological and ecological drought events according to drought severity".

  RESPONSE: We agree and can see your suggestion. The caption of this section has been revised to "Top ten meteorological and ecological drought events according to drought severity".

- For Figure 5, as the double y axes are used, I suggest the authors display them with different colors, e.g., the red and blue y-axes are used to display the extent of area and severity. Similarly, please revise Figure 8.

  RESPONSE: Thanks for the hint. We have revised Figure 5 and Figure 8 according to your suggestion.

[Figure]

Figure 5: Temporal evolution of DS and DA of (a) meteorological drought event No. 87 and

(b) ecological drought event No. 127.

[Figure]

Figure 8: Sensitivity test of overlapping areas of drought patches between two adjacent

months.

- Line 255-256, the "of five-fold cross-validations" should be removed from the latter part of this sentence as the relevant statement has been presented in the former.

  RESPONSE: Thanks for the hint. We have deleted the "of five-fold cross-validations" in the latter part of this sentence.

- Line 257-258, the GP and MP should be listed as full names when they appear for the first time.

  RESPONSE: Thanks for the hint. We have corrected it to " Most models showed good performance except for Gaussian Process and Multi-layer Perceptron."

- In Figures 2 and 7, based on the terms commonly used, the drought levels regarding the "serious" should be revised as "severe". Of course, the same statement about this need to be changed throughout the manuscript.

  RESPONSE: Thanks for your advice, we have revised "serious" to "severe" in Figure 2, Figure 7, and related content throughout the manuscript.

[Figure]

Figure 2: A schematic diagram illustrating the procedure of the drought propagation identification method.

[Figure]

Figure 7: Conditional probability of ecological drought at (a) extreme, (b) severe, and (c) moderate levels, given that characteristics of meteorological drought exceed a certain value.

- In Figure 2, I think the title in purple color should be changed to "Constructing response model of ecological drought to meteorological drought".

RESPONSE: Thanks, we have changed it according to your suggestion.

[Figure]

Figure 2: A schematic diagram illustrating the procedure of the drought propagation identification method.

- In the caption of Figure 7, the "different levels" should be pointed out to increase the

readability, e.g., (a) extreme, (b) severe, and (c) moderate. Please check it.

RESPONSE: Thanks, we have changed it according to your suggestion.

MODIFICATION:

[Figure]

Figure 7: Conditional probability of ecological drought at (a) extreme, (b) severe, and (c) moderate levels, given that characteristics of meteorological drought exceed a certain value.

References:

Jehanzaib, M., Sattar, M. N., Lee, J.-H., and Kim, T.-W.: Investigating effect of climate change on drought propagation from meteorological to hydrological drought using multi-model ensemble projections, STOCHASTIC ENVIRONMENTAL RESEARCH AND RISK ASSESSMENT, 34, 7–21, https://doi.org/10.1007/s00477-019-01760-5, 2020.

Xu, Y., Zhang, X., Hao, Z., Singh, V. P., and Hao, F.: Characterization of agricultural drought propagation over China based on bivariate probabilistic quantification, JOURNAL OF HYDROLOGY, 598, https://doi.org/10.1016/j.jhydrol.2021.126194, 2021.

---

## Author Comment (AC4)

Dear Referee #1:

Thank you for your kind and encouraging comments on our study. Your comments and our responses to them are listed below.

- Powerful machine learning approaches were applied. What were the degrees of freedom of the machine learning approaches? What is the ratio of the degrees of freedom over the rather small number of 81 meteorological drought events?

RESPONSE: It seems these two problems were posed in order to investigate whether more degrees of freedom would cause the model to overfit the training data. In general, Regularization techniques and optimal model architectures are employed to ensure machine learning models are not overfitted and maintain low generalization errors. Therefore, degrees of freedom and model complexity always correspond very poorly (Janson et al., 2013), which is generally much less than the number of parameters in the model (Gao and Jojic, 2016). In this study, we used a Python package called PyCaret to construct these classifiers. L2 regularization method was selected in each model to avoid overfitting and maintain high calculation efficiency. The relevant description will be added to Section 2.4.3 in **Line 197-Line 204**.

MODIFICATION: In this study, each binary classifier was constructed using a Python package called PyCaret, which wraps several machine-learning libraries, including scikit-learn, XGBoost, LightGBM, CatBoost, spaCy, Optuna, and Hyperopt (Ali, 2020). The tune_model() function in the PyCaret package offers simple selection of optimal hyperparameters of each model. A 5-fold cross-validation was used to train and validate the classifiers in each model by setting "fold=5" in the create_model() function. In using the compare_models() function, the classifier with the highest summation of accuracy, precision, recall, F1 score, and Matthews correlation coefficient was selected as the optimal model. To avoid overfitting and maintain high calculation efficiency, the L2 regularization method was selected for each model by setting the parameter "penalty='l2'".

- According to Fig. 7 propagation probability is nearly exclusively determined by the

severity of the meteorological drought which would meet common expectations. In contrast, any effect of duration or area is hardly discernible. Please compare the performance of the machine learning approaches to that of a multivariate linear regression

RESPONSE: We agree and can see your point. In this study, machine learning models were used to determine whether a meteorological drought event has propagating potential. It is therefore a binary classification question. We have added Figure 9 and a description related to it in Section 3.4 in the revised version.

MODIFICATION:

[Figure]

Figure 9 Three-dimensional diagram showing characteristics of meteorological drought events. Larger circles indicate greater severity.

As can be seen in Figure 9, propagated meteorological droughts have greater severity, larger affected area, and longer duration than non-propagated droughts.

RESPONSE: We agree that any effect of duration or area is hardly discernible. In this study, meteorological drought and ecological drought with genetic relationship were extracted on the basis of a certain spatio-temporal matching rule. Therefore, the model constructed in this study only includes meteorological drought and ecological drought events that have genetic relationships. As a result, only 103 out of 184 ecological drought events were

induced by 81 out of 108 meteorological drought events. The severity of ecological drought thus can be predicted based on the characteristics of meteorological drought. We have added the relevant description below in Section 4.1 in the revised version.

MODIFICATION: Using this method, two types of drought events without spatial connection would be excluded (only 103 out of 184 ecological drought events were induced by 81 out of 108 meteorological drought events), and more drought characteristics, such as affected area and migration path could be extracted. This addresses the limited applicability of the traditional method to regions with large spatial extent, and provides more reliable information for quantifying relationship between characteristics of meteorological drought and ecological drought.

RESPONSE: We have included your recommendation and added multivariable linear regression in **Line 317-Line 324** in the revised version.

MODIFICATION: For comparison, ternary linear and ternary quadratic models were constructed based on 46 pairs of meteorological-ecological drought events (Table 8). The comparisons were made in terms of three independent variables, M_DS, M_DD, and M_DA, and one dependent variable, E_DS. As shown in Table 8, the $R^2$ of the ternary quadratic model was evidently higher than that of the ternary linear model, whereas the RMSE, AIC, and BIC were lower. This illustrates that M_DS, M_DD, M_DA, and E_DS follow a nonlinear relationship, and that the ternary quadratic model is more suitable for simulating their relationship. According to the ternary quadratic model, E_DS equals $1.4\times 10^6$ month·km$^2$ when M_DA > $17.6\times10^5$ km$^2$ $\cap$ M_DD > 11.8 month $\cap$ M_DS > $7.5\times 10^6$ month·km$^2$. These values correspond to the thresholds of moderate ($1.7\times 10^6$ month·km$^2$), severe ($2.4\times10^6$ month·km$^2$), and extreme ($4.6\times10^6$ month·km$^2$) ecological drought.

**Table 8 Modelling E_DS with polynomial functions based on meteorological drought characteristics**

| Model types | Expression | Assessment metrics | | | |
|---|---|---|---|---|---|
| | | RMSE | AIC | BIC | $R^2$ |
| Ternary linear model | $E\_DS=4.85\times10^5+0.15M\_DS+4099.35M\_DD-1.20M\_DA$ | $9.24\times10^5$ | 1350.67 | 1357.89 | 0.58 |
| Ternary quadratic model | $E\_DS=1.54-0.05M\_DS-16.91M\_DD-0.08M\_DA-1319.23M\_DD^2+0.03M\_DD\times M\_DA$ | $7.29\times10^5$ | 1085.75 | 1100.20 | 0.85 |

- Please check the use of definite and indefinite articles and the use of plural "s".

RESPONSE: Thanks for the hint. We have checked them carefully to avoid grammar errors in the revised version and marked them with red color.

Details:

- 53-55: Who is "they"?

RESPONSE: Thanks for the hint. The sentence "In other words, they considered temporal connection of two drought types and ignored their spatial overlap, which may result in the miscalculation of drought propagation in regions with large spatial extent." has been changed to " In other words, the traditional statistical methods only consider the temporal connection between two drought types and ignore their spatial overlap, which may result in the miscalculation of drought propagation in regions with large spatial extent." **in Line 52-Line 54.**

- 73-79: Section "2 Study area" comprises only 6 lines and should be merged with the subsequent section 3, or at least with section "3.1 Datasets".

RESPONSE: We agree. Section 2 has been changed to Section 2.1 in the revised version.

- 82-85: Verb is missing.

RESPONSE: Thanks for the hint. The sentence " Monthly meteorological data, including surface reflectance, temperature, relative humidity, atmospheric pressure, downward shortwave radiation, wind speed, and longwave radiation, obtained from the ERA5-land reanalysis dataset (https://cds.climate.copernicus.eu) issued by European Centre for

Medium-Range Weather Forecasts (ECWMF), which has a spatial resolution of $0.1° × 0.1°$ and covers the period of 1981–2021" has been changed to " Monthly meteorological data, including surface reflectance, temperature, relative humidity, atmospheric pressure, downward shortwave radiation, wind speed, and longwave radiation, was obtained from the ERA5-land reanalysis dataset (https://cds.climate.copernicus.eu) issued by European Centre for Medium-Range Weather Forecasts (ECWMF), which has a spatial resolution of $0.1° × 0.1°$ and covers the period of 1981–2021." in **Line 86-Line 89.**

- 91: Use lowercase letter in "Root".

RESPONSE: Thanks for the hint. We have corrected it to "root" in **Line 95**.

- 98: Replace "deep phreatic buried depth" by "great depth to groundwater".

RESPONSE: Thanks for the hint. We have corrected it to " great depth to groundwater " in **Line 101.**

- 112: Both "SEWDI" and "SEBS" need to be explained in a concise way. Referring to the Jiang et al. (2021) paper does not suffice.

RESPONSE: We agree and can see your suggestion. We have explained it in **Line 114-Line 120** in the revised version.

MODIFICATION: SEWDI follows a similar procedure as SPI, which includes the calculation of ecological water deficit (EWD), the selection of an optimal distribution for fitting monthly EWD series, and the inverse normal transformation of the cumulative density distribution of EWD. EWD is the difference between ecological water requirement (EWR) and ecological water consumption (EWC) (Chi et al., 2018; Jiang et al., 2021). Among them, EWR was calculated using the single crop coefficient method recommended by the Food and Agriculture Organization (FAO). EWC equals the actual evapotranspiration, which is derived from latent heat fluxes calculated by the surface energy balance system (SEBS) algorithm.

- 124: Should be "three steps", not "two steps".

RESPONSE: Thanks for the hint. We have corrected it to " three steps " in **Line 132**.

- 147: Delete "to".

RESPONSE: Thanks for the hint. We have deleted "to".

- 200: Do you mean "Johnson S_B distribution"?

RESPONSE: Thanks for the hint. We have corrected "johnsonsb" to "Johnson S_B" in the full text.

- 224: What does "DS" mean?

RESPONSE: DS represents drought severity. We have changed the name to its full form.

- 265: Please explain "itau method".

RESPONSE: We have added " The Copula estimation can be eased by the itau method, which inverts Kendall's tau method (Demarta and McNeil, 2005)." in **Line 302-Line 303** for explaining "itau method".

- 280-297: Section 5.1 should be either part of the methods or of the results section.

RESPONSE: Thanks for your suggestion, we have moved Section 5.1 to the results section (Section 3.1) in the revised version.

- 349-352: Verb is missing.

RESPONSE: Thanks for the hint. The verb has been added to this sentence.

MODIFICATION: Monthly meteorological data, including surface reflectance, temperature, relative humidity, atmospheric pressure, downward shortwave radiation, wind speed, and longwave radiation, was obtained from the ERA5-land reanalysis dataset (https://cds.climate.copernicus.eu) issued by European Centre for Medium-Range Weather Forecasts (ECWMF), which has a spatial resolution of $0.1° × 0.1°$ and covers the period of 1981–2021.

- Figure 3: I guess that the drought event numbers reflect chronological order, is that right? The colour scale indicates about the same meteorological-ecological drought event number for very different ecological and meteorological drought event numbers. E.g., green symbols show up for ecological drought event number 1-10, 30-50 and >150. How can that be? Is there something wrong with the colour coding of the symbols?

RESPONSE: You are right, we have corrected this mistake. Figure 3 has been replaced with the figure below in Figure 4 in the revised version.

[Figure]

Figure 4: Identification results of paired meteorological and ecological drought events

- Figure 7: In the figure caption correct "exceeding" to "exceed".

RESPONSE: Thanks for the hint. We have corrected "exceeding" to "exceed".

References:

Ali, M.: PyCaret,   PyCaret: An open source, low-code machine learning library in Python, 2020.

Demarta, S. and McNeil, A. J.: The t Copula and Related Copulas, International Statistical Review / Revue Internationale de Statistique, 73, 111–129, 2005.

Gao, T. and Jojic, V.: Degrees of Freedom in Deep Neural Networks, https://doi.org/10.48550/arXiv.1603.09260, 2016.

Janson, L., Fithian, W., and Hastie, T.: Effective Degrees of Freedom: A Flawed Metaphor, https://doi.org/10.1093/biomet/asv019, 2013.

---

## Author Comment (AC5)

Dear Referee #2:

First of all, thank you very much for the helpful comments and your kind feedback, your comments are listed below, along with our responses to each one.

- Some works, e.g., the systematic literature review of propagation probability model, need to be added into the Introduction or Discussion sections.

RESPONSE: Thanks for your advice. The purpose of this section is to introduce the use of Bayesian model to explore probabilistic relationships among different types. We have rewritten this section based on your advice. You can find it in **Line 55-Line 69**.

MODIFICATION: The probability information of one type of successive drought events is contained in another type of associated drought (Wu et al., 2021). Therefore, a number of studies have attempted to assess the propagation relationships between the two drought types based on the probabilistic method. A Bayesian network is a probabilistic model that acquires probabilistic inferences over interacting variables of interest based on a graphical structure. Therefore, this method has been proven to be suitable for quantifying the probability relationship between different drought types (Ayantobo et al., 2018; Chang et al., 2016; Das et al., 2020). For example, Guo et al. (2020) calculated the occurrence probability of hydrological drought based on different intervals of duration and severities of meteorological drought. Sattar et al (2019) identified the occurrence probability of different classes and lag times of hydrological drought according to the intensity of meteorological drought. Xu et al. (2021) found that the probability of agricultural drought severity increased synchronously with meteorological drought in different regions of China. Jehanzaib et al. (2020) concluded that in the Korean Peninsula, the probability of meteorological drought propagating into hydrological drought increased significantly under climate change. In general, these studies primarily focused on the relationship between duration and severity between the two drought types but ignored the relationships among affected areas. Xu et al. (2015a) found that the probability of drought occurrence would be underestimated if drought affected areas are not considered. Therefore, the traditional probabilistic model of

drought propagation can be improved by introducing the three-dimensional clustering method, which would provide more drought information (Liu et al., 2019).

- Line 214 (i.e., Eq. (10)), the "n – 1" is shown in the inner product should be revised as "n – i". Moreover, please define or explain the symbols that appeared in this and some other equations, e.g., the term c is not defined. Please carefully check them.

RESPONSE: Thanks for your hint. We indeed made a mistake when typing Eq.(10). We have corrected it and added the explanation of the symbols. Furthermore, all equations have been carefully checked to ensure no errors would occur. The modification is located in **Line 227-Line 230**.

MODIFICATION:

$$f(x_1,\ldots,x_n) = \prod_{i=1}^{n} f_i(x_i) \times \prod_{i=1}^{n-1} \prod_{j=1}^{n-i} c_{i,i+j|1:(i-1)} \left\{ F(x_i \mid x_1,\ldots,x_{i-1}), F(x_{i+j} \mid x_1,\ldots,x_{i-1}) \right\} \tag{10}$$

where $f(x_1,\ldots,x_n)$ represents the joint density function. $c$ represents bivariate Copula densities, which includes Gumbel, Gaussian, Frank, and Clayton Copula function; $F$ represents cumulative distribution function of marginal distribution. $i$ and $j$ represent root nodes.

- For Section 4.2, please provide specific information about paired drought events so that we can identify the characteristics of four paired categories. At the same time, I found the statement "Among them, the peaks of the meteorological drought event appeared two months ahead (December 2007) that of ecological drought (February 2007)" may be wrong in Lines 250-251. As you know, the duration between December 2007 and February 2007 is far more than two months. Please revise it.

RESPONSE: This is a good suggestion. We have provided specific information about paired drought events to show the characteristics of four paired categories. The relevant description will be added in Section 3.3 in **Line 276-Line 279**.

MODIFICATION:   Meteorological drought of OMT type showed a longer duration, a larger affected area, and a greater severity than ecological drought. However, this is contrary to

type MTO. Simultaneously, ecological drought of type MTO showed a longer duration, a larger affected area, and a greater severity than those of type OTM (Figure 5).

[Figure]

Figure 5. A box plot showing the intensity, duration, and affected area of paired meteorological-ecological drought among different types

MODIFICATION: The statement "Among them, the peaks of the meteorological drought event appeared two months ahead (December 2007) that of ecological drought (February 2007)" was corrected to " Among them, the peaks of the meteorological drought event appeared two months ahead (December 2007) that of ecological drought (February 2008)." in **Line 285-Line 286**.

- An essential part of this article is the use of machine learning to solve a binary classification problem. In this context, I suggest adding a plot that shows the severity, duration, and affected area of meteorological droughts propagated and didn't propagate.

RESPONSE: That is an excellent suggestion. Thank you very much for that as well! In this study, machine learning models were used to determine whether a meteorological drought event has propagating potential. We have added Figure 9 and related descriptions in Section 3.4 with reference to the advice of Referee#1.

MODIFICATION:

[Figure]

Figure 9 Three-dimensional diagram showing characteristics of meteorological drought events. Larger circles indicate greater severity.

As can be seen in Figure 9, propagated meteorological droughts have greater severity, larger affected area, and longer duration than non-propagated droughts.

- I suggest replacing "Three-dimensional drought identification method" with "Three-dimensional clustering method".

RESPONSE: We agree and can see your point. We will replace "Three-dimensional drought identification method" with "Three-dimensional clustering method" in the text.

Details:

● Lines 24-25, for specification, the drought classification should be changed as the "drought types". Please revise it.

RESPONSE: Thanks for the hint. We have corrected "drought classification" to "drought types" in **Line 26**.

● Lines 66-68, "Taking a typically ecological fragile region … to meteorological drought", I suggest revising it as "Taking a typically ecological fragile region, Northwestern China (NWC), as an example, the motivation of this study, from a three-dimensional perspective, is proposed a novel hybrid machine learning-Copula method to investigate the response probability of ecological drought to meteorological drought" to highlight the novelty of this paper.

RESPONSE: Thanks. We have followed your advice and changed it.

● Line 94-99, authors should refine their explanation of why SPI-3 is used to represent meteorological drought.

RESPONSE: Thanks for the hint. Line 94-99 has been changed to " Previous studies found that the standardized precipitation evaporation index (SPEI) overestimated the meteorological drought in NWC where actual atmospheric water demand is determined by precipitation variation (Ayantobo and Wei, 2019; Zhang et al., 2019a; Zhang et al., 2021b). Additionally, precipitation is the main water resources for vegetation growth in most regions of NWC due to the great depth to groundwater (Cao et al., 2021). Standardized precipitation index (SPI) was thus used in the current study to represent meteorological drought. SPI at different time scales was calculated by aggregating $n$-month moving sums, allowing the identification of various drought types (McKee et al., 1993). At short time scales, drought events are characterized by high frequency and short duration, while at long time scales, they have longer duration and lower frequency. SPI–3 has been reported to be highly representative of the impacts of meteorological conditions on vegetation as vegetation variation is sensitive to precipitation accumulated over three months (McKee et

al., 1993; Vicente-Serrano et al., 2012; Vicente-Serrano et al., 2010). Therefore, SPI-3 was used to characterize meteorological drought in this study." **in Line 102-Line 107.**

- Section 3.3.2, the number of steps regarding the Spatiotemporal connection of two drought types may be disordered, e.g., the statements of steps were listed as Firstly and Secondly in Lines 154-155, but that remained as the Secondly in Line 163. Please check it.

  RESPONSE: Thanks for the hint. We will revise "Secondly" in line 163 in original manuscript to "Thirdly" in **Line 170** in the revision version.

- Lines 206-207, please revised the "Cramer-von Mises (CM) test" as the "Cramer-von Mises (CvM) test" based on common sense.

  RESPONSE: Thanks for the hint. We have corrected it to "Cramer-von Mises (CvM) test" in **Line 220.**

- Line 224, I recommended the authors revised the caption of this section as "Top ten meteorological and ecological drought events according to drought severity".

  RESPONSE: We agree and can see your suggestion. The caption of this section has been revised to "Top ten meteorological and ecological drought events according to drought severity" **in Line 256.**

- For Figure 5, as the double y axes are used, I suggest the authors display them with different colors, e.g., the red and blue y-axes are used to display the extent of area and severity. Similarly, please revise Figure 8.

  RESPONSE: Thanks for the hint. We have revised Figure 5 and Figure 8 according to your suggestion. Now, they have been changed to Figure 7 and Figure 3 in the revision version.

[Figure]

Figure 7: Temporal evolution of DS and DA of (a) meteorological drought event No. 87 and

(b) ecological drought event No. 127.

[Figure]

Figure 3: Sensitivity test of overlapping areas of drought patches between two adjacent

months.

● Line 255-256, the "of five-fold cross-validations" should be removed from the latter part of this sentence as the relevant statement has been presented in the former.

RESPONSE: Thanks for the hint. We have deleted the "of five-fold cross-validations" in the latter part of this sentence.

● Line 257-258, the GP and MP should be listed as full names when they appear for the first time.

RESPONSE: Thanks for the hint. We have corrected it to " Most models showed good performance except for Gaussian Process and Multi-layer Perceptron." in **Line 294**.

● In Figures 2 and 7, based on the terms commonly used, the drought levels regarding the "serious" should be revised as "severe". Of course, the same statement about this need to be changed throughout the manuscript.

RESPONSE: Thanks for your advice, we have revised "serious" to "severe" in Figure 2, Figure 7 (it has been changed to Figure 10 in the revision version), and related content throughout the manuscript.

[Figure]

Figure 2: A schematic diagram illustrating the procedure of the drought propagation identification method.

[Figure]

Figure 10: Conditional probability of ecological drought at (a) extreme, (b) severe, and (c) moderate levels, given that characteristics of meteorological drought exceed a certain value.

- In Figure 2, I think the title in purple color should be changed to "Constructing response model of ecological drought to meteorological drought".

  RESPONSE: Thanks, we have changed it according to your suggestion.

[Figure]

Figure 2: A schematic diagram illustrating the procedure of the drought propagation identification method.

- In the caption of Figure 7, the "different levels" should be pointed out to increase the

readability, e.g., (a) extreme, (b) severe, and (c) moderate. Please check it.

RESPONSE: Thanks, we have changed it according to your suggestion.

MODIFICATION:

[Figure]

Figure 7: Conditional probability of ecological drought at (a) extreme, (b) severe, and (c) moderate levels, given that characteristics of meteorological drought exceed a certain value.

References:

Jehanzaib, M., Sattar, M. N., Lee, J.-H., and Kim, T.-W.: Investigating effect of climate change on drought propagation from meteorological to hydrological drought using multi-model ensemble projections, STOCHASTIC ENVIRONMENTAL RESEARCH AND RISK ASSESSMENT, 34, 7–21, https://doi.org/10.1007/s00477-019-01760-5, 2020.

Xu, Y., Zhang, X., Hao, Z., Singh, V. P., and Hao, F.: Characterization of agricultural drought propagation over China based on bivariate probabilistic quantification, JOURNAL OF HYDROLOGY, 598, https://doi.org/10.1016/j.jhydrol.2021.126194, 2021.

---

## Author Response (AR2)

We would like to thank the editor again for the constructive feedback and comments. We addressed the remaining questions of the editor and updated our responses. The editor's comments are written in black. Our responses are written in blue.

- Can you please specify the meaning of MCC in your equation 9?

RESPONSE: Thanks for the hint. We have added the description of equation 9 in **Line 153-155 in the Track-change files.**

MODIFICATION: MCC represents Matthew's correlation coefficient, which is used to evaluate the accuracy of binary classification tasks.

- You mentioned P1 in section 2.4, but this marginal probability is not mentioned again. Can you please clarify your notation?

RESPONSE: Thanks for the hint. Originally, we planned to use "P1" as an abbreviation, but it is not mentioned again in the remainder of the text. So, we have deleted it in **line 140 and line 145-146 in the Track-change file.**

MODIFICATION: The probabilities of ecological drought at different magnitudes triggered by meteorological drought were obtained by multiplying their conditional probability with the propagated probability of meteorological drought.

- Is there something missing after Step 3?. If not include a dot at the end of the sentence.

RESPONSE: Thanks for the hint. We have added a dot at the end of the sentence.

- There is a missing equal sign in equation 11

RESPONSE: Thanks for the hint. The format of Eq.11 has been changed to avoid misunderstandings as it lacks an equal sign. Equation 11 is in **Line 237-240 in the Track-changed file.**

MODIFICATION:

$$F(X>x\,|\,D>d,A>a,S>s) = \frac{F(D>d,S>s,A>a,X>x)}{F(S>s,A>a,D>d)}$$
$$= (1 - F(d) - F(s) - F(a) - F(x) + C(F_D(d),F_S(s))$$
$$+ C(F_D(d),F_A(a)) + C(F_D(d),F_X(x)) + C(F_A(a),F_S(s))$$
$$+ C(F_A(a),F_X(x)) + C(F_X(x),F_S(s)) - C(F_D(d),F_S(s),F_A(a))$$
$$- C(F_D(d),F_S(s),F_X(x)) - C(F_D(d),F_A(s),F_X(x))$$
$$- C(F_S(s),F_A(a),F_X(x)) + C(F_D(d),F_S(s),F_A(a),F_X(x)))$$
$$/(1 - F_D(d) - F_A(a) - F_S(s) + C(F_D(d),F_A(a))$$
$$+ C(F_D(d),F_S(s)) + C(F_A(a),F_S(s)) - C(F_D(d),F_A(a),F_S(s))) \qquad (11)$$

where *D*, *A*, and *S* represent duration, area, and severity of propagated meteorological drought, respectively; *X* represents ecological drought at moderate, severe, and extreme magnitudes, which equals the cumulative probability of 0.5, 0.7, and 0.9, respectively. *C* represents the cumulative distribution function of the joint distribution.

- Can you please define the meaning of MTO and OTM in Line 290?

RESPONSE: We are sorry that the relevant abbreviations were not been unified. Actually, MTO and OTM represent many-to-one and one-to-many respectively in Line 179 to 185. We have checked the full text to ensure that the abbreviations are uniform.

MODIFICATION: The sentence " Meteorological drought of OTM type showed a longer duration, a larger affected area, and a greater severity than ecological drought. However, this is contrary to type MTO. Simultaneously, ecological drought of type MTO showed a longer duration, a larger affected area, and a greater severity than those of type OTM (Figure 5)" has been changed to "Meteorological drought of type one-to-many showed a longer duration, a larger affected area, and a greater severity than ecological drought. However, this is contrary to type many-to-one. Simultaneously, ecological drought of type many-to-one showed a longer duration, a larger affected area, and a greater severity than those of type one-to-many (Figure 5)" in Line 280-283 in the Track-changed file.

- Figure 2: a) In the text you use P1, but in the figure you use P(1). Please clarify the notation. B) In the top title of your diagram located in the bottom left of the figure it reads:" Constructing response model of ecological drought to meteorological drought". But the diagram and your method suggest you should use: Meteorological drought to Ecological drought. Please clarify.

RESPONSE: a) Thanks for the hint. The p1 in Figure 2 has been deleted since we abandoned its use in section 2.4.1. In addition, "probability" has been added. b) Thanks for your suggestion, we have changed "Constructing response model of ecological drought to meteorological drought" to "Constructing propagation model of meteorological drought to ecological drought".

[Figure]

- Lines 550 and 555: Please revise unusual symbols in the references

RESPONSE: Thanks for the hint. We have corrected references in Line 550 and 555 to "

Thom, H. C. S.: A Note on the Gamma Distribution, Monthly Weather Review, 86, 117-122, 10.1175/1520-0493, 1958." and "Vicente-Serrano, S. M., Begueria, S., Lorenzo-Lacruz, J., Camarero, J. s. J., Lopez-Moreno, J. I., Azorin-Molina, C., Revuelto, J. s., Moren-Tejeda, E., and Sanchez-Lorenzo, A.: Performance of Drought Indices for Ecological, Agricultural, and Hydrological Applications, Earth Interactions, 16, 1-27, 10.1175/2012ei000434.1, 2012." in Line 536 and 544 in the Track-changed file.